# Kernel Multimodal Continuous Attention

**Alexander Moreno**[*]
Luminous Computing

**Zhenke Wu**
University of Michigan

**Supriya Nagesh**
Georgia Tech

**Walter Dempsey**
University of Michigan

**James M. Rehg**
Georgia Tech

## Abstract

Attention mechanisms average a data representation with respect to probability weights. Recently, [23–25] proposed continuous attention, focusing on unimodal exponential and deformed exponential family attention densities: the latter can have sparse support. [8] extended to multimodality via Gaussian mixture attention densities. In this paper, we propose using kernel exponential families [4] and our new sparse counterpart, kernel *deformed* exponential families. Theoretically, we show new existence results for both families, and approximation capabilities for the deformed case. Lacking closed form expressions for the context vector, we use numerical integration: we prove exponential convergence for both families. Experiments show that kernel continuous attention often outperforms unimodal continuous attention, and the sparse variant tends to highlight time series peaks.

## 1 Introduction

Attention mechanisms [3] are weighted averages of data representations used to make predictions. Discrete attention 1) cannot easily handle irregularly spaced observations, and 2) attention maps may be scattered, lacking focus. [23, 24] proposed continuous attention, showing that attention densities maximize the regularized expectation of a function of the data location (i.e. time). Special cases lead to exponential and deformed exponential families: the latter has sparse support. They form a continuous data representation and take expectations with respect to attention densities. In [25] they apply this to a transformer architecture.

[23–25] used unimodal attention densities, giving importance to *one* data region. [8] extended this to multimodal Gaussian mixture attention densities. However 1) Gaussian mixtures lie in neither the exponential nor deformed exponential families, and are difficult to study in the context of [23, 24]; and 2) they have dense support. Sparse support can say that certain regions of data do not matter: a region of time has *no* effect on class probabilities, or a region of an image is *not* some object. We would like to use multimodal exponential and deformed exponential family attention densities, and understand how [8] relates to the framework of [23, 24].

This paper makes three contributions: 1) we introduce kernel *deformed* exponential families, a sparse multimodal density class, and apply it along with the multimodal kernel exponential families [4] as attention densities. The latter have been used for density estimation, but not weighting data importance; 2) we theoretically analyze kernel exponential and deformed exponential family i) normalization, ii) approximation and iii) context vector numerical integration properties; 3) we apply

---

[*]An earlier draft was done at Georgia Tech as a dissertation chapter. Additional experiments and extensive rewriting were done at Luminous Computing.

36th Conference on Neural Information Processing Systems (NeurIPS 2022).

them to real world datasets, showing that multimodal continuous attention outperforms unimodal, and that kernel deformed exponential family densities often highlight the peaks of time series. Approximation properties for the kernel deformed case are challenging: similar kernel exponential family results [33] relied on exponential and logarithm properties to bound the difference of the log-partition functional at two functions: these do not hold for deformed analogues. We provide similar bounds by using a mean value inequality and bounding a functional derivative.

We first review unimodal continuous attention [23, 24]. We motivate multimodal continuous attention via time warping. We next describe kernel exponential families and give a novel normalization condition relating kernel growth to the base density's tail decay. We then propose kernel deformed exponential families, new densities with support over potentially disjoint regions. We describe normalization and approximation capabilities. Next we describe using these densities for continuous attention, including numerical integration convergence analysis. We show experiments comparing unimodal and multimodal attention, and conclude with limitations and future work.

## 2 Related Work

**Attention Mechanisms** closely related are [23–25, 8]. [23, 24] frame continuous attention as an expectation of a value function with respect to a density, where the density solves an optimization problem. They only used unimodal (deformed) exponential family densities: we extend this to the multimodal setting by leveraging kernel exponential families and proposing a deformed counterpart. [8] proposed a multimodal continuous attention mechanism via a Gaussian mixture. We show in Appendix A that this solves a slightly different optimization problem from [23, 24]. A limitation of Gaussian mixtures is lack of flexible tail decay. Finally, [25] apply continuous attention within a transformer architecture to model long context. This is a new application of continuous attention rather than an extension of specific continuous attention mechanisms.

Also relevant are [40, 30, 31]. [30] provide an attention mechanism for irregularly sampled time series by use of a continuous-time kernel regression framework, but do not take an expectation of a data representation over time with respect to a continuous pdf. Instead they evaluate the kernel regression model at fixed time points. This describes importance of data at a set of points rather than over continuous regions. [31] extend this to incorporate uncertainty quantification. Other papers connect attention and kernels, but focus on discrete attention [40, 5]. Also relevant are temporal transformer papers, including [45, 15, 17, 32]. However, none have continuous attention densities.

**Kernel Exponential Families** [4] proposed kernel exponential families: [33] analyzed theory for density estimation. [44] parametrized the kernel with a deep neural network. Other density estimation papers include [1, 6, 36]. We apply kernel exponential families as attention densities to *weight* a value function which represents the data, rather than for density estimation. Further, [44] showed a condition for an unnormalized kernel exponential family density to have a finite normalizer. However, they used exponential power base densities. We instead relate kernel growth rates to the base density tail decay, allowing non-symmetric base densities.

To summarize our theoretical contributions: 1) showing that multimodal continuous attention is required to represent time warping 2) introducing kernel *deformed* exponential families with approximation and normalization analysis 3) improved kernel exponential family normalization results 4) stability and convergence analysis of numerical integration for kernel-based continuous attention 5) characterizing [8] in terms of the framework of [23, 24].

## 3 Continuous Attention Mechanisms

An attention mechanism has: 1) a value function: a raw or learned data representation 2) an attention density chosen to be 'similar' to another data representation, encoding it into a density 3) a context $c$ [23] taking an expectation of the value function with respect to the attention density:

$$c = \mathbb{E}_{T \sim p}[V(T)]. \tag{1}$$

The value function $V : S \to \mathbb{R}^D$ is a data representation, $T \sim p(t)$ is the random variable or vector for locations (temporal, spatial, etc) in domain $S$, and $p(t)$ is the attention density (potentially with respect to a discrete measure). For discrete attention, one could have $V(t)$ be a time series where $t \in S$ a finite set of time points. One then weights the time series with a probability vector to

obtain the context vector. An example of this for irregularly sampled time series is [30]. For each attention mechanism (eqn. 3 in their paper), $S$ is the set of observed time points, the value function maps the observed time points to their respective observation values, and the attention probability mass function is the output of normalized kernel evaluations between observation time points and a reference time point.

In the continuous setting $V(t)$ could be a curve or realization of a continuous-time stochastic process, and $S$ could be $[0, \tau]$ where $\tau$ is a study end time. One then weights it with a continuous probability density. If $S$ is a set of spatial locations and one has image data, then the value function could be a raw image or learned representation of an image. Finally, for self-attention [18, 42], $V$ is a linear transformation of a sequence, and the expectation is conditional on a transformation of a specific token (query).

To choose $p$, one takes a data representation $f$ and finds $p$ 'similar' to $f$, but regularizing $p$. [23, 24] did this, formalizing attention mechanisms. Given a measure space $(S, \mathcal{A}, Q)$, let $\mathcal{M}_+^1(S)$ be the set of probability densities with respect to $Q$. Assume $Q$ is dominated by a $\sigma$-finite measure $\nu$ (i.e. Lebesgue) and that it has Radon Nikodym derivative $q_0 = \frac{dQ}{d\nu}$ with respect to $\nu$. Let $S \subseteq \mathbb{R}^D$, $\mathcal{F}$ be a function class, and $\Omega : \mathcal{M}_+^1(S) \to \mathbb{R}$ be a lower semi-continuous, proper, strictly convex functional. Given $f \in \mathcal{F}$, an *attention density* [23] $\hat{p} : S \to \mathbb{R}_{\geq 0}$ solves

$$\hat{p}[f] = \arg \max_{p \in \mathcal{M}_+^1(S)} \int_S p(t)f(t)dQ(t) - \Omega(p). \tag{2}$$

This maximizes regularized $L^2$ similarity between $p$ and a data representation $f$. If $\Omega(p) = \int_S p(t) \log p(t) dQ(t)$ is the negative differential entropy, the attention density is Boltzmann Gibbs

$$\hat{p}[f](t) = \exp(f(t) - A(f)), \tag{3}$$

where $A(f)$ ensures $\int_S \hat{p}[f](t)dQ = 1$ (see [23] for proof). If $f(t) = \theta^T \phi(t)$ for parameters and statistics $\theta \in \mathbb{R}^M, \phi(t) \in \mathbb{R}^M$ respectively, Eqn. 3 becomes an exponential family density. For $f$ in a reproducing kernel Hilbert space (RKHS) $\mathcal{H}$, it becomes a kernel exponential family density [4], which we propose as an alternative attention density.

One desirable class would be heavy or thin tailed exponential family like densities. In exponential families, the support, or non-zero region of the density, is controlled by the measure $Q$. Letting $\Omega(p)$ be the $\alpha$-Tsallis negative entropy $\Omega_\alpha(p)$ [41],

$$\Omega_\alpha(p) = \begin{cases} \frac{1}{\alpha(\alpha-1)} \left( \int_S p(t)^\alpha dQ - 1 \right), \alpha \neq 1; \\ \int_S p(t) \log p(t) dQ, \alpha = 1, \end{cases}$$

then $\hat{p}[f]$ for $f(t) = \theta^T \phi(t)$ lies in the deformed exponential family [41, 27]

$$\hat{p}_{\Omega_\alpha}[f](t) = \exp_{2-\alpha}(\theta^T \phi(t) - A_\alpha(f)), \tag{4}$$

where $A_\alpha(f)$ again ensures normalization and the density uses the $\beta$-exponential

$$\exp_\beta(t) = \begin{cases} [1 + (1 - \beta)t]_+^{1/(1-\beta)}, \beta \neq 1; \\ \exp(t), \beta = 1. \end{cases} \tag{5}$$

For $\beta < 1$, Eqn. 5 and thus deformed exponential family densities for $1 < \alpha \leq 2$ can return 0 values. Values $\alpha > 1$ (and thus $\beta < 1$) give thinner tails than the exponential family, while $\alpha < 1$ gives fatter tails. Setting $\beta = 0$ is called *sparsemax* [22]. In this paper, we assume $1 < \alpha \leq 2$, which is the sparse case studied in [23]. We again propose to replace $f(t) = \theta^T \phi(t)$ with $f \in \mathcal{H}$, which leads to the novel *kernel deformed exponential families*.

Computing Eqn. 1's context vector requires parametrizing $V(t)$. [23] parametrize $V : S \to \mathbb{R}^D$ with $\mathbf{B} \in \mathbb{R}^{D \times N}$ as $V(t; \mathbf{B}) = \mathbf{B}\Psi(t)$ and estimate $\mathbf{B} \in \mathbb{R}^{D \times N}$ via regularized multivariate linear regression. Here $\Psi = \{\psi_n\}_{n=1}^N$ is a set of basis functions. Let $L$ be the number of observation locations (times in a temporal setting), $D$ be the observation dimension, and $N$ be the number of basis functions. Using Frobenius norm $\| \cdot \|_F$, this involves regressing the observation matrix $\mathbf{H} \in \mathbb{R}^{D \times L}$ on a matrix $\mathbf{F} \in \mathbb{R}^{N \times L}$ of basis functions $\{\psi_n\}_{n=1}^N$ evaluated at observation locations $\{t_l\}_{l=1}^L$

$$\mathbf{B}^* = \arg \min_{\mathbf{B}} \|\mathbf{B}\mathbf{F} - \mathbf{H}\|_F^2 + \lambda \|\mathbf{B}\|_F^2, \tag{6}$$

## 4 Time Warping

We now draw a connection to time warping [28] to show an advantage of our method. One summary statistic for classification is a global weighted average pooling: an expectation of temporal features. However in many processes features may not be aligned in time, and we only observe unaligned curves due to a discrepancy between individual system times and clock time: this is known as phase variation [21]. For instance, electrocardiogram (ECG) heartbeat curves have a P-wave, a QRS complex and a T-wave. These have similar patterns between heartbeats, but may have different durations and peak locations. Here we show that the expectation of a temporally aligned curve with respect to a global density is equivalent to the expectation of the unaligned curve with respect to an individualized density. However even if the global density is unimodal, the individualized density may not be. We first define the function that aligns a set of features to common reference times.

**Definition 4.1.** (Time Warping Function) Given references times $\{t_{0k}\}_{k=1}^{K}$ and individualized times $\{t_{ik}\}_{k=1}^{K}$, both in $[0, \tau]$, a **time warping function** $h_i : S \to \mathbb{R}$ for $S \subseteq \mathbb{R}_{\geq 0}$ is a strictly increasing, differentiable, invertible function where

$$h_i(0) = 0, h_i(\tau) = \tau$$
$$h_i(t_{0k}) = t_{ik}, k = 1, \ldots, K$$
$$h_i(t) = t \text{ if } t \notin [0, \tau]$$

Let $\{X_i\}_{i=1}^{n}, X_i : S \to \mathbb{R}$ be observed curves, each with $K$ features occurring at individualized times $\{t_{ik}\}_{k=1}^{K} \subset [0, \tau]$ increasing in $k$. A set of time warping functions $\{h_i\}_{i=1}^{n}$ map reference times to individualized feature times. One can then compute aligned $X_i^*(t) = X_i(h_i(t))$. Each $X_i^*$ has relevant features at the same times $\{t_{0k}\}_{k=1}^{K}$. Classically, this requires handcrafting and locating important features and estimating a warping function. We could then compute an expectation of the time warped curve with respect to a *global* fixed density $p(t)$ to obtain a summary statistic $\mathbb{E}_{T \sim p} X_i^*(T)$ of the aligned curve. The following states that multimodal continuous attention can represent such an expectation with an attention density $p_i$, avoiding computing $X_i^*(t)$.

**Lemma 4.2.** *(Continuous Attention can Represent Time Warping) Let $h$ be a time warping function, $g = h^{-1}$ and $X_i : \mathbb{R} \to \mathbb{R}$ with support on $[0, \tau]$. Assume that $Q$ is dominated by Lebesgue measure $\nu$ and let $q_0 = \frac{dQ}{d\nu}$. Then for any fixed density $p$ wrt $Q$, if $g, X_i, q_0, p$ are continuous almost everywhere we have*

$$\mathbb{E}_{U \sim p} X_i^*(U) = \mathbb{E}_{T \sim p_i} X_i(T) \tag{7}$$

*where $p_i(t) = p(g_i(t)) \frac{q_0(g_i(t))}{q_0(t)} g_i'(t)$ and $p_i(t)$ is a valid probability density function.*

See Appendix B.1 for proof. Even if $p(t)$ is unimodal, $p_i(t)$ may not be: see Appendix B.2 for an example. Thus we require multimodal continuous attention to represent such statistics.

## 5 Kernel Exponential and Deformed Exponential Families

We use kernel exponential families and a new deformed counterpart to obtain flexible attention densities solving Eqn. 2 with the same regularizers. We first review kernel exponential families. We then give a novel theoretical result describing when an unnormalized kernel exponential family density can be normalized. This says that the normalizing constant exists when the base density has fast enough tail decay relative to kernel growth. This result allows us to verify that a choice of base measure and kernel lead to a valid attention density and thus attention mechanism. Next we introduce kernel deformed exponential families, extending kernel exponential families to have either sparse support, our focus, or fatter tails. These can attend to non-overlapping time intervals. We show similar normalization results based on kernel choice and base density. The normalizing constant exists when the unnormalized density has compact support and the kernel grows sufficiently slowly. Following this we show approximation theory. We conclude by showing how to compute attention densities in practice.

Kernel exponential families [4] extend exponential families, replacing $f(t) = \theta^T \phi(t)$ with $f \in \mathcal{H}$ a reproducing kernel Hilbert space $\mathcal{H}$ [2]. Densities can be written

$$p(t) = \exp(f(t) - A(f))$$
$$= \exp(\langle f, k(\cdot, t) \rangle_{\mathcal{H}} - A(f)). \tag{8}$$

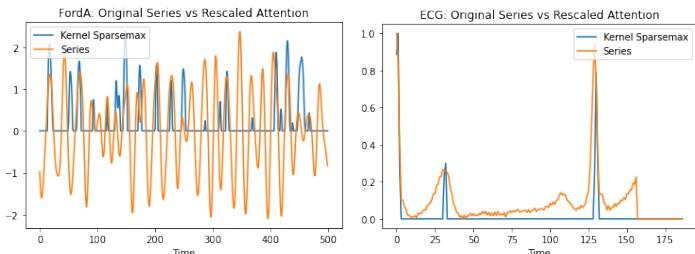

Figure 1: Rescaled continuous attention densities for a) kernel deformed exponential families on an engine noise example b) kernel deformed exponential families on an ECG example. Both examples are multimodal and highlight salient features of a signal, and the ECG example in particular highlights the waves, which describe the electrical signal passing through the heart conduction system. It first highlights the R wave, the largest peak, then the T wave, and next the R wave again.

Eqn. 8 follows from the reproducing property. A challenge is to choose $\mathcal{H}, Q$ so that a normalizing constant exists, i.e., $\int_S \exp(f(t))dQ < \infty$. These densities can approximate any continuous density over a compact domain arbitrarily well in KL divergence, Hellinger, and $L^p$ distance [33]. However relevant integrals including the normalizing constant require numerical integration.

To avoid infinite dimensionality one generally assumes a representation of the form $f = \sum_{i=1}^I \gamma_i k(\cdot, t_i)$, where for density estimation [33] the $t_i$ are the observation locations and this is the solution to a regularized empirical risk minimization problem. This requires using one parameter per observation value. This model complexity may not be necessary, and often one chooses a set of *inducing points* [38] $\{t_i\}_{i=1}^I$ where $I$ is less than the number of observation locations.

For a given pair $\mathcal{H}, k$, how can we choose $Q$ to ensure that the normalization constant exists? We first give a simple example of $\mathcal{H}, f$ and $Q$ where it *does not*.

*Example* 1. Let $Q$ be the law of a $\mathcal{N}(0,1)$ distribution and $S = \mathbb{R}$. Let $\mathcal{H} = \text{span}\{t^3, t^4\}$ with $k(t,s) = t^3 s^3 + t^4 s^4$ and $f(t) = t^3 + t^4 = k(t,1)$. Then the following integral diverges.

$$\int_S \exp(f(t))dQ = \int_{\mathbb{R}} \frac{1}{\sqrt{2\pi}} \exp\left(-\frac{t^2}{2} + t^3 + t^4\right) dt$$

## 5.1 Theory for Kernel Exponential Families

We provide sufficient conditions for $Q$ and $\mathcal{H}$ so that $A(f)$ the log-partition functional exists. We relate $\mathcal{H}$'s kernel growth rate to the tail decay of the random variable or vector $T_Q$ with law $Q$.

**Proposition 5.1.** *Let $\tilde{p}(t) = \exp(f(t))$ where $f \in \mathcal{H}$ an RKHS with kernel $k$. Assume $k(t,t) \leq L_k\|t\|_2^\xi + C_k$ for constants $L_k, C_k, \xi > 0$. Let $Q$ be the law of a random vector $T_Q$, so that $Q(A) = P(T_Q \in A)$. Assume $\forall u$ s.t. $\|u\|_2 = 1, z > 0$*

$$P(|u^T T_Q| \geq z) \leq C_q \exp(-vz^\eta) \tag{9}$$

*for some constants $\eta > \frac{\xi}{2}, C_Q, v > 0$. Then*

$$\int_S \tilde{p}(t)dQ < \infty.$$

See Appendix C.1 for proof. Based on $k(t,t)$'s growth, we can vary what tail decay rate for $T_Q$ ensures we can normalize $\tilde{p}(t)$. [44] also proved normalization conditions, but focused on exponential power density for a specific growth rate of $k(t,t)$ rather than relating tail decay to growth rate. By focusing on tail decay, our result can be applied to non-symmetric base densities. Specific kernel bound growth rate terms $\xi$ lead to allowing different tail decay rates.

**Corollary 5.2.** *For $\xi = 4$, $T_Q$ can be any sub-Gaussian random vector. For $\xi = 2$ it can be any sub-exponential. For $\xi = 0$ it can be any probability density.*

See Appendix C.2 for proof.

## 5.2 Kernel Deformed Exponential Families

We now propose kernel deformed exponential families, which are flexible sparse non-parametric densities: these can be multimodal. They take deformed exponential families and extend them to use kernels in the deformed exponential term. This mirrors kernel exponential families. We write

$$p(t) = \exp_{2-\alpha}(f(t) - A_\alpha(f)),$$

where $f \in \mathcal{H}$ with kernel $k$. Fig. 1 shows that they can have support over disjoint intervals.

### 5.2.1 Normalization Theory

We construct a valid kernel deformed exponential family density from $Q$ and $f \in \mathcal{H}$. We first discuss the deformed log-normalizer. In exponential family densities, the log-normalizer is the log of the normalizer. For deformed exponentials, the following holds.

**Lemma 5.3.** *Let $Z > 0$ be a constant. Then for $1 < \alpha \leq 2$,*

$$\frac{1}{Z} \exp_{2-\alpha}(Z^{\alpha-1} f(t)) = \exp_{2-\alpha}(f(t) - \log_\alpha Z)$$

*where*

$$\log_\beta t = \begin{cases} \frac{t^{1-\beta}-1}{1-\beta} & \text{if } t > 0, \beta \neq 1; \\ \log(t) & \text{if } t > 0, \beta = 1; \\ \text{undefined} & \text{if } t \leq 0. \end{cases}$$

See Appendix D.1 for proof. We now describe a normalization sufficient condition analagous to Proposition 5.1 for deformed kernel exponential families. With Lemma 5.3 and an unnormalized $\exp_{2-\alpha}(\tilde{f}(t))$ we derive a valid normalized kernel deformed exponential family density. We only require that an affine function of the terms in the deformed-exponential is negative for large $\|t\|_2$.

**Proposition 5.4.** *For $1 < \alpha \leq 2$ assume $\tilde{p}(t) = \exp_{2-\alpha}(\tilde{f}(t))$ with $\tilde{f} \in \mathcal{H}$, $\mathcal{H}$ is a RKHS with kernel $k$. If $\exists C_t > 0$ s.t. for $\|t\|_2 > C_t$, $(\alpha - 1)\tilde{f}(t) + 1 \leq 0$ and $k(t,t) \leq L_k\|t\|_2^\xi + C_k$ for some $\xi > 0$, then $\int_S \exp_{2-\alpha}(\tilde{f}(t))dQ < \infty$.*

See Appendix D.2 for proof. We now construct a valid kernel deformed exponential family density using the finite integral.

**Corollary 5.5.** *Under proposition 5.4's conditions, assume $\exp_{2-\alpha}(\tilde{f}(t)) > 0$ on a set $A \subseteq S$ where $Q(A) > 0$, then $\exists$ constants $Z > 0$, $A_\alpha(f) \in \mathbb{R}$ such that for $f(t) = \frac{1}{Z^{\alpha-1}}\tilde{f}(t)$, the following holds*

$$\int_S \exp_{2-\alpha}(f(t) - A_\alpha(f))dQ = 1.$$

See Appendix D.3 for proof. We thus estimate $\tilde{f}(t) = Z^{\alpha-1}f(t)$ and normalize to obtain a density of the desired form.

### 5.2.2 Approximation Theory

Kernel deformed exponential families can approximate continuous densities satisfying a tail condition on compact domains arbitrarily well in $L^p$ norm, Hellinger distance, and Bregman divergence.

**Theorem 5.6.** *Let $q_0 \in C(S)$, such that $q_0(t) > 0$ for all $t \in S$, where $S \subseteq \mathbb{R}^d$ is locally compact Hausdorff and $q_0$ is the Radon Nikodym derivative of measure $Q$ with respect to a dominating measure $\nu$. Suppose there exists $l > 0$ such that for any $\epsilon > 0$, $\exists R > 0$ satisfying $|p(t) - l| \leq \epsilon$ for any $t$ with $\|t\|_2 > R$. Define*

$$\mathcal{P}_c = \{p \in C(S) : \int_S p(t)dQ = 1, p(t) \geq 0, \forall t \in S \text{ and } p - l \in C_0(S)\}.$$

*Suppose $k(t, \cdot) \in C_0(S) \forall t \in S$ and the kernel integration condition (Eqn. 12) holds. Then kernel deformed exponential families are dense in $\mathcal{P}_c$ wrt $L^r$ norm, Hellinger distance and Bregman divergence for the $\alpha$-Tsallis negative entropy functional.*

The proof (Appendix D.4) idea is that under a kernel integrability condition, deformed exponential families parametrized by $f \in \mathcal{H}$ are dense in those parametrized by $f \in C_0(S)$ [2] (we denote those parametrized by $f \in C_0(S)$ as $\mathcal{P}_0$). We can approximate $C(S)$ densities satisfying the tail condition with $\mathcal{P}_0$ densities, and thus with deformed exponential family densities. This extends [33]'s approximation to the deformed case: standard log and exponential rules cannot be applied. It requires bounding functional derivatives and applying a mean value inequality.

## 5.3 Using Kernels for Continuous Attention

Here we describe how to compute continuous attention mechanisms with attention densities parametrized by functions in an RKHS $\mathcal{H}$ in practice. Algorithm 1 shows the kernel deformed exponential family case: the kernel exponential family case involves a similar algorithm. Given a base measure, kernel, and inducing point locations, we start by computing kernel weights $\tilde{\gamma}_i$ for $\tilde{f}(t) = Z^{\alpha-1}f(t) = \sum_{i=1}^{I} \tilde{\gamma}_i k(t, t_i)$ and estimating the matrix $\mathbf{B}$ for basis weights for the value function $V(t) = \mathbf{B}\Psi(t)$. Unlike density estimation, this form for $f$ is simply a practical way to obtain $f$ in an RKHS, rather than a solution to an empirical risk minimization problem. We then compute the normalizing constant $Z = \int_S \exp_{2-\alpha}(\tilde{f}(t))dQ$ via numerical integration and use it to normalize $\tilde{f}(t)$ to obtain the attention density $p(t)$. Finally we compute the context $c = \mathbb{E}_{T \sim p}[V(T)] = \mathbf{B}\mathbb{E}_p[\Psi(t)]$ by taking the expectation of $\Psi(T)$ with respect to a deformed kernel exponential family density $p$. Unlike [23, 24], we lack closed form expressions and use numerical integration. In the backwards pass we use automatic differentiation. Note that in some cases we have numerical underflow when computing the normalizing constant. We also found that using FP16 precision, the default on some newer GPUs for Pytorch up to 1.11, leads to worse performance, sometimes dramatically so, and thus recommend using FP32. We discuss solutions for the underflow issue in Appendix E.1.

---

**Algorithm 1** Continuous Attention Mechanism via Kernel Deformed Exponential Families

---

**Choose** $q_0(t)$ and kernel $k$. Inducing point locations $\{t_i\}_{i=1}^{I}$
**Parameters** $\{\tilde{\gamma}_i\}_{i=1}^{I}$ the weights for $\tilde{f}(t) = (Z)^{\alpha-1}f(t) = \sum_{i=1}^{I} \tilde{\gamma}_i k(t, t_i)$, matrix $\mathbf{B}$ for basis weights for value function $V(t) = \mathbf{B}\Psi(t)$. $I$ is number of inducing points.
**Forward Pass**
Compute $Z = \int \exp_{2-\alpha}(\tilde{f}(t))dQ(t)$ to obtain $p(t) = \frac{1}{Z}\exp_{2-\alpha}(\tilde{f}(t))$ via numerical integration
Compute $\mathbb{E}_{T \sim p}[\Psi(T)]$ via numerical integration
Compute $c = \mathbb{E}_{T \sim p}[V(T)] = \mathbf{B}\mathbb{E}_p[\Psi(T)]$
**Backwards Pass** use automatic differentiation

---

### 5.3.1 Numerical Integration Convergence

The trapezoidal rule's standard one-dimensional convergence rate is $O(\frac{1}{N^2})$ for an integral over a fixed interval, where $N$ is the number of grid points. We would like better convergence guarantees. We can achieve exponential convergence for the numerical integrals of kernel exponential and deformed exponential family attention. We focus on numerical integration over the real line, leaving truncation analysis and higher dimensions to future work. We let $h > 0$ be the grid size.

For functions holomorphic in a strip with rapid decay, the trapezoidal rule has exponential convergence. For kernel exponential family attention, this gives us $O(\exp(-C/h))$ or exponential convergence for some $C > 0$ with appropriate choice of $q_0$, $V$, and $k$. Technical details are in E.2, and are based on extending real-valued analytic functions to complex functions analytic/holomorphic on a strip.

Kernel deformed exponential families, however, are not even differentiable, but we can construct a sequence of differentiable approximations by replacing the positive part/ReLU function in the deformed exponential with softplus for increasing values of the softplus parameter. Each differentiable approximation has exponential convergence, and by taking limits as the softplus parameter tends to infinity we can show that the numerical integral for kernel deformed exponential family attention itself has exponential convergence. Technical details are in E.3.

We also show empirical convergence analysis in Appendix E.4 and figure 2 in that appendix. Both kernel exponential and deformed exponential families see rapid convergence for 1d attention, pro-

---

[2]continuous function on domain $S$ vanishing at infinity

| Attention | Accuracy |
|---|---|
| Cts Softmax | 77.72±14.20 |
| Cts Sparsemax | 77.96±9.64 |
| Kernel Softmax (ours) | **85.71 ± 11.98** |
| Kernel Sparsemax (ours) | **88.1 ± 1.50** |

Table 1: Results for 100 runs each of synthetic time warping classification experiment, $N = 64$. This involves generating $10,000$ trajectories, each of length 95 of unaligned curves.

viding excellent integral approximations with only 5-10 grid points. Further, the numerical integral using softplus is a very close approximation to that using ReLU for softplus parameters 5 and 10.

## 6   Experiments

We investigate how often multimodal continuous attention outperforms unimodal, given the same architecture. We also investigate whether these methods learn rich multimodal densties. We denote kernel exponential family attention as kernel softmax and the deformed case as kernel sparsemax. Our architectures have: 1) an encoder maps a discrete time series representation to attention density parameters. 2) The value function $V(t; \mathbf{B})$ expresses an embedding of a time series as a linear transformation of basis functions. 3) The context is $c = \mathbb{E}_p[V(T)]$, which is used in 4) a classifier. Fig. 3 in the Appendices visualizes this. For kernel softmax/sparsemax the encoder outputs are the weights for the kernel evaluations. For the Gaussian mixture case, the encoder's outputs are the mixture weights and components means and variances. Here we describe one synthetic and three real data experiments. In Appendix E we provide an additional ECG classification experiment. We provide confidence intervals for our three smaller experiments (at most $10,000$ samples). Our UWave and ECG experiments were done on a Titan X GPU, IMDB on a 1080, and FordA on an A40. We found that the A40 provides very different results out of the box for both accuracy/F1 and attention densities. As an example, Figure 8 was done on a Titan X with an older version of Pytorch, while Figure 7 was done with an A40 with Pytorch 1.12. Code is in our repository[3], where we discuss the flags used to control precision on recent GPUs and Pytorch versions. We discuss computational/ memory complexity in Appendix F, and give a summary in Table 5. We provide wall clock times in Table 6.

### 6.1   Synthetic Experiment: Time Warping

We simulate time warping and do prediction for unimodal vs multimodal attention densities. Details of the original time aligned stochastic process and inverse warping function are in Appendix B.3. Given global densities for classes $p_1, p_2$ and aligned $X^*$, the class is $\arg\max(\langle p_1, X^* \rangle, \langle p_2, X^* \rangle)$. We generate $10,000$ trajectories of length 95 observed at evenly spaced time points in the interval $[0, 1]$. We use two attention mechanisms (heads), one for each class. Letting $V$ be the value function fit to observed data, the classifier is $\text{softmax}([\langle p_{i1}, V \rangle_{L^2}, \langle p_{i2}V \rangle_{L^2}])$ where $p_{i1}$ and $p_{i2}$ are the attention densities for the first and second class. Table 1 shows prediction results from 100 runs along with 1.96 standard deviation intervals. Kernel methods outperform by $7 - 10\%$.

### 6.2   FordA Dataset

This is a binary classification dataset for whether a sympton exists in an automotive subsystem. Each time series has 500 sensor observations, and there are 3601 training samples and 1320 test samples. Table 2 shows results. Kernel sparsemax outperforms most baselines[4], while kernel softmax also does well. We use the same methods and architecture as the previous section, although hyperparameters are slightly different. Several methods, including discrete softmax, continuous sparsemax and the transformer, have very poor performance. To sanity check we fit SVM with a Gaussian kernel, logistic regression, and a decision tree: accuracies are under 55% for these, and we conclude that

---

[3]https://github.com/onenoc/kernel-continuous-attention

[4]An earlier pre-print of this paper using previous TSAI/PyTorch versions and a Titan X instead of an A40 had $\sim 90\%$ accuracy/F1 for LSTM FCN. We no longer have access to that environment, but we suspect the difference has to do with either precision or implementation changes in the TSAI library.

| Method | Accuracy | F1 |
|---|---|---|
| Discrete Softmax | $51.27 \pm 1.86$ | $33.89 \pm 0.82$ |
| Cts Softmax | $74.88 \pm 21.36$ | $74.56 \pm 21.84$ |
| Cts Sparsemax | $90.95 \pm 1.77$ | $90.94 \pm 1.77$ |
| Gaussian Mixture | $69.56 \pm 35.94$ | $60.75 \pm 52.88$ |
| Kernel Softmax (ours) | $\mathbf{92.44 \pm 1.96}$ | $\mathbf{92.43 \pm 2.00}$ |
| Kernel Sparsemax (ours) | $\mathbf{92.61 \pm 1.62}$ | $\mathbf{92.60 \pm 1.62}$ |
| LSTM FCN | $\mathbf{93.41 \pm 0.43}$ | $\mathbf{93.40 \pm 0.43}$ |
| TST (Transformer) | $49.48 \pm 1.04$ | $49.46 \pm 1.02$ |

Table 2: FordA accuracy, 1.96 SD intervals over 10 runs. Our methods outperform most baselines. Several methods have *very* poor performance. To sanity check we fit three classical methods: kernel SVM, logistic regression, and a decision tree. All have under 55% accuracy, suggesting that this dataset is problematic for some methods.

| Attention | N=64 | N=128 |
|---|---|---|
| Cts Softmax | $67.78 \pm 1.64$ | $67.70 \pm 2.49$ |
| Cts Sparsemax | $74.20 \pm 2.72$ | $74.69 \pm 3.78$ |
| Gaussian Mixture | $81.13 \pm 1.76$ | $80.99 \pm 2.79$ |
| Kernel Softmax (ours) | $\mathbf{93.85 \pm 0.60}$ | $\mathbf{94.26 \pm 0.75}$ |
| Kernel Sparsemax (ours) | $\mathbf{92.32 \pm 1.09}$ | $\mathbf{92.15 \pm 0.79}$ |

Table 3: Accuracy results on uWave gesture classification dataset for the irregularly sampled case. Again over 10 runs. Due to the irregular sampling, this is only comparable to [16, 29]. Kernel based attention substantially outperforms unimodal and mixture models. All methods use 100 attention heads. Gaussian mixture uses 100 components (and thus 300 parameters per head), and kernel methods use 256 inducing points.

this data poses difficulty for some classifiers. Appendix G provides some additional details, along with attention density plots. The kernel softmax plots often highlight zero crossings. The kernel sparsemax plots often select peaks of a signal while learning rich sparsity patterns.

### 6.3 uWave Experiment: Gesture Classification

We investigate: 1) Does a large number of unimodal attention heads, as suggested in [23, 24], perform well when multimodality is needed? 2) Can this method work well for irregularly sampled time series? 3) Can we learn interesting multimodal attention densities?

We analyze uWave [19]: accelerometer time series with eight gesture classes. We follow [16]'s split into 3,582 training observations and 896 test observations: sequences have length 945. We do synthetic irregular sampling and uniformly sample 10% of the observations. Because of this our results are comparable to other uWave irregular sampling papers [16, 29], but *not* to results using the full time series.

Table 3 shows the results. Our highest accuracy is 94.26%, the multi-head unimodal case's best is 74.69%, and the mixture's best is 81.13%. We outperform the results of [16], who report a highest accuracy of 91.41%, and perform similarly to [29] (their figure suggests approximately 94% accuracy). Fig. 10 shows attention densities for one of the attention heads for the first three classes. This takes one attention density for each time series of each class and plots it. Within the same class, all attention densities for the head (one for each time series) are plotted. The plot shows two things: firstly, attention densities have support over non-overlapping time intervals. This cannot be done with Gaussian mixtures, and the intervals would be the same for each density in the exponential family case. Secondly, there is high similarity of attention densities within each class, but low similarity between classes. Appendix H describes additional details.

| Attention | N=32 | N=64 | N=128 | Mean |
|---|---|---|---|---|
| Cts Softmax | 89.56 | 90.32 | **91.08** | 90.32 |
| Cts Sparsemax | 89.50 | 90.39 | 89.96 | 89.95 |
| Kernel Softmax | **91.30** | **91.08** | 90.44 | **90.94** |
| Kernel Sparsemax | 90.56 | 90.20 | 90.41 | 90.39 |

Table 4: IMDB sentiment classification dataset accuracy. Continuous softmax uses Gaussian attention, continuous sparsemax truncated parabola, and kernel softmax and sparsemax use kernel exponential and deformed exponential family with a Gaussian kernel. The latter has $\alpha = 2$ in exponential and multiplication terms. $N$: basis functions, $I = 10$ inducing points, bandwidth 0.01.

### 6.4 IMBD Sentiment Classification

We extend [23]'s code[5] for IMDB sentiment classification [20]. This uses a document representation $v$ from a convolutional network and an LSTM attention model. We use a Gaussian base density and kernel, and divide the interval $[0, 1]$ into $I = 10$ inducing points where we evaluate the kernel in $f(t) = \sum_{i=1}^{I} \gamma_i k(t, t_i)$. Table 4 shows results. On average, kernel exponential and deformed exponential family slightly outperforms the continuous softmax and sparsemax. The continuous softmax/sparsemax results are from running their code.

## 7 Discussion

In this paper we extend continuous attention mechanisms to use kernel exponential and deformed exponential family densities. The latter is a new flexible class of non-parametric densities with sparse support. We show novel existence properties for kernel exponential and deformed exponential families, prove approximation properties for the latter, and show exponential convergence of numerical integration for both attention mechanisms. We then apply these to the continuous attention framework described in [23, 24]. We show results on several datasets. Kernel attention mechanisms tend to outperform unimodal attention, sometimes by a large margin. We also see that in many cases they exhibit multimodality. Kernel sparsemax in particular learns rich sparsity patterns while highlighting peaks of a signal. This was evident in Figure 1. The engine noise example mimicked certain parts of the signal, while in the ECG example, it tends to give very high weight to R waves in a signal. While this paper is more focused on general methods than a specific potentially dangerous application, potential application areas include wearable sensors and NLP, and general negative societal impacts in those application areas could apply to this work.

### 7.1 Limitations

A limitation was the use of numerical integration, which scales poorly with location dimensionality. While we achieve 1D exponential convergence, we still must investigate whether we can extend this to higher dimensions. This still allows for multiple observation dimensions at a given 1d location, i.e. multivariate time series or language tasks.

## 8 Funding and Other Acknowledgments

Alexander Moreno is supported by Luminous Computing, and was previously supported by NIH 1-P41-EB028242-01A1, U01CA229437, and a Google CMD-IT Flip-Alliance fellowship. Zhenke Wu is partly supported by NIH U01CA229437 and an investigator grant from Precision Health Initiative at University of Michigan, Ann Arbor. Supriya Nagesh is supported by NSF CNS1823201. Walter Dempsey is supported by P50 DA054039. We thank Marcos Treviso and André Martins for helpful discussions about their prior work and help with understanding their code.

---

[5][23]'s repository for this dataset is https://github.com/deep-spin/quati

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
