# Appendix

## Table of Contents

## Overview of Appendices

Appendices A to E are mostly theoretical. In Appendix A we describe how the Gaussian mixture attention from [8] relates to the theoretical framework of [23, 24]. In Appendix B we discuss time warping, showing that sufficiently flexible continuous attention can represent time warping, that in at least in one case a multimodal density is required, and give some details describing our experiments. In Appendix C we discuss normalization for kernel exponential families, while in Appendix D we do the same along with approximation results for kernel deformed exponential families. In Appendix E we discuss rate of convergence for approximating the attention mechanism with numerical integration, and show some synthetic experiments showing rapid convergence empirically.

The remaining appendices are empirical. In Appendix H, we include an additional experiment analyzing an accelerometer classification dataset, where we add synthetic irregular sampling. In Appendix I we provide additional details for the MIT-BIH experiment. Finally, in Appendix G we provide additional details for the FordA experiment. The latter two appendices in particular focus on visual comparisons of attention densities. We find that kernel deformed exponential families/sparsemax tend to learn more interesting looking attention densities than all of the other methods, and for ECG kernel sparsemax has the interpretation of selecting regions where the electrical stimuli are largest.

The licenses are: for MIT BIH open data commons attribution. For UCR time series (uWave and Ford) the exact license is difficult to find but the citation is [7]. For IMDB it is again difficult to find the exact license, but the original paper is [20] and the website[6] asks that it be cited.

## A  Gaussian Mixture Model

In this section we descibe how the Gaussian mixture attention of [8] relates to the optimization definition of attention densities in [23, 24]. In fact their attention densities solve a related but different optimization problem. [23, 24] show that exponential family attention densities maximize a regularized linear predictor of the expected sufficient statistics of locations. In contrast, [8] find a joint density over locations and latent states, and maximize a regularized linear predictor of the expected *joint* sufficient statistics. They then take the marginal location densities to be the attention densities.

Let $\Omega(p)$ be Shannon entropy and consider two optimization problems:

$$\arg \max_{p \in \mathcal{M}_+^1(S)} \langle \theta, \mathbb{E}_p[\phi(T)] \rangle_{l^2} - \Omega(p)$$

$$\arg \max_{p \in \mathcal{M}_+^1(S)} \langle \theta, \mathbb{E}_p[\phi(T, Z)] \rangle_{l^2} - \Omega(p)$$

The first is Eqn. 2 with $f = \theta^T \phi(t)$ and rewritten to emphasize expected sufficient statistics. If one solves the second with variables $Z$, we recover an Exponential family joint density

$$\hat{p}_{\Omega_\alpha}[f](t, z) = \exp(\theta^T \phi(t, z) - A(\theta)).$$

This encourages the joint density of $T, Z$ to be similar to a *complete data* representation $\theta^T \phi(t, z)$ of both location variables $T$ and latent variables $Z$, instead of encouraging the density of $T$ to be similar to an observed data representation $\theta^T \phi(t)$. Let $S_Z$ be the domain for latent variables $Z$. The latter optimization is equivalent to

$$\arg \max_{p \in \mathcal{M}_+^1(S \times S_Z)} \Omega(p)$$

$$\text{s.t.}$$

$$\mathbb{E}_{p(T,Z)}[\phi_m(T, Z)] = c_m, m = 1, \ldots, M.$$

The constraint terms $c_m$ are determined by $\theta$. Thus, this maximizes the joint entropy of $Z$ and $T$, subject to constraints on the expected joint sufficient statistics.

To recover EM learned Gaussian mixture densities, one must select $\phi_m$ so that the marginal distribution of $T$ will be a mixture of Gaussians, and relate $c_m$ to the EM algorithm used to learn the mixture model parameters. For the first, assume that $Z$ is a multinomial random variable with a single trial taking $|Z|$ possible values and let $\phi(t, z) = (I(z = 1), \ldots, I(z = |Z| - 1), I(z = 1)t, I(z = 1)t^2, \ldots, I(z = |Z|)t, I(z = |Z|)t^2)$. These are multinomial sufficient statistics with a single trial, followed by the sufficient statistics of $|Z|$ Gaussians multiplied by indicators for each $z$. Then $p(T|Z)$ will be Gaussian, $p(Z)$ will be multinomial, and $p(T)$ will be a Gaussian mixture. For constraints, [8] have

$$\mathbb{E}_{p(T,Z)}[\phi_m(T, Z)] = \sum_{l=1}^{L} w_l \sum_{k=1}^{|Z|} p_{\text{old}}(Z = k | T = t_l) \phi_m(t_l, k), \tag{10}$$

$$m = 1, \ldots, M$$

[6]http://ai.stanford.edu/ amaas/data/sentiment/

at each EM iteration: $p_{\text{old}}(Z|T = t_l)$ is the previous iteration's latent state density conditional on the observation location, $w_l$ are discrete attention weights, and $t_l$ is a discrete attention location. That EM has this constraint was shown in [43]. This matches expected joint sufficient statistics to those implied by discrete attention over locations, taking into account the dependence between $Z$ and $T$ given by old model parameters. An alternative is simply to let $\theta$ be the output of a neural network. While the constraints lack the intuition of Eqn. 10, it avoids the need to run EM. We focus on this case and use it for our baselines.

## B    Time Warping

### B.1    Proof of 4.2

Here we prove that continuous attention can represent time warping. We change measure to Lebesgue, use the equivalence between Lebesgue and Riemann for almost everywhere continuous functions, use integration by substitution, and then change back.

$$
\begin{aligned}
\mathbb{E}_p X_i^*(U) &= \int_{[0,\tau]} p(u) X_i^*(u) dQ \\
&= \int_{[0,\tau]} p(u) q_0(u) X_i^*(u) d\nu(u) \\
&= \int_0^\tau p(u) q_0(u) X_i^*(u) du \\
&= \int_0^\tau p(g_i(t)) q_0(g_i(t)) X_i^*(g_i(t)) g_i'(t) dt \\
&= \int_0^\tau p(g_i(t)) \frac{q_0(g_i(t))}{q_0(t)} g_i'(t) X_i^*(g_i(t)) q_0(t) dt \\
&= \int_{[0,\tau]} p(g_i(t)) \frac{q_0(g_i(t))}{q_0(t)} g_i'(t) X_i^*(g_i(t)) dQ \\
&= \int_{[0,\tau]} p_i(t) X_i(t) dQ
\end{aligned}
$$

Further, since $h$ is strictly increasing, $g$ is as well so that $g_i'(t) > 0 \forall t \in [0, \tau]$. This gives us that $p_i(t)$ is non-negative, and we have to show that it integrates to 1.

$$
\begin{aligned}
\int_{[0,\tau]} p(g_i(t)) \frac{q_0(g_i(t))}{q_0(t)} g_i'(t) dQ &= \int_0^\tau p(g_i(t)) q_0(g_i(t)) g_i'(t) dt \\
&= \int_0^\tau p(u) q_0(u) du \\
&= 1
\end{aligned}
$$

### B.2    Multimodal Densities are Required

Consider the case where $p_{\text{global}}(t)$ is an exponential density function and $g(t)$ the inverse warping function has the following form

$$
p_{\text{global}}(t) = \begin{cases} \lambda \exp(-\lambda(t)), t \geq 0 \\ 0, \text{ else} \end{cases}
$$

$$
g(t) = \begin{cases} C_1 \int_0^t \exp(W(u)) du, t \in [0, \tau] \\ t \text{ else} \end{cases}
$$

for $C_1 = \frac{\tau}{\int_0^\tau \exp(W(u))du}$ and some twice diferentiable function $W$. The role of $C_1$ is to ensure that $g(\tau) = \tau$. Then

$$p_i(t) = \lambda C_1 \exp(-\lambda C_1 \int_0^t \exp(W(u))du + W(t))$$

so that

$$p_i'(t) = (-\lambda C_1 \exp(t) + W'(t))p_i(t)$$
$$p_i''(t) = (-\lambda C_1 \exp(t) + W''(t))p_i(t)$$
$$+ (-\lambda C_1 \exp(t) + W'(t))^2 p_i(t)$$

Now note that depending on the sign and magnitude of $W''(t)$, $p_i''(t)$ may be either positive or negative and thus $p_i(t)$ may lack a unique optimum and may be multimodal.

### B.3 Time Warping Experimental Details

This describes the aligned vs the observed stochastic process. The original time aligned stochastic process is given by

$$X^*(t) = \begin{cases} Z_0 \cos 9\pi t, 0 \le t < 0.25 \\ Z_1 t^2, 0.25 \le t < 0.5 \\ Z_2 \sin t, 0.5 \le t < 0.75 \\ Z_3 \cos 17\pi t, 0.75 \le t < 1 \end{cases}$$

where $Z_k \sim U(-4, 4), k = 1, 2, 3, 4$. We instead observe realizations $X_i(t) = X_i^*(g(t))$ at fixed time points with the following inverse warp function

$$g_i(t) = \begin{cases} C_i \int_0^t \exp(-s\lambda_i)ds, \lambda_i \sim U(0, 25), t \in [0, 1] \\ t \text{ else} \end{cases}$$

The global densities are $p_1 \sim U(0, 0.5)$ and $p_2 \sim U(0.5, 1)$.

## C Proof Related to Proposition 5.1

### C.1 Proof of Proposition 5.1

Here we prove that if the kernel evaluated twice at the same point grows sufficiently slowly with respect to base density tail decay, then we can normalize.

*Proof.* This proof has several parts. We first bound the RKHS function $f$ and use the general tail bound we assumed to give a tail bound for the one dimensional marginals $T_{Qd}$ of $T_Q$. Using the RKHS function bound, we then bound the integral of the unnormalized density in terms of expectations with respect to these finite dimensional marginals. We then express these expectations over finite dimensional marginals as infinite series of integrals. For each integral within the infinite series, we use the finite dimensional marginal tail bound to bound it, and then use the ratio test to show that the infinite series converges. This gives us that the original unnormalized density has a finite integral.

We first note, following [44], that the bound on the kernel in the assumption allows us to bound $f$ in terms of two constants and the absolute value of the point at which it is evaluated.

$$|f(t)| = |\langle f, k(t, \cdot) \rangle_\mathcal{H}| \text{ reproducing property}$$
$$\le \|f\|_\mathcal{H} \|k(t, \cdot)\|_\mathcal{H} \text{ Cauchy Schwarz}$$
$$= \|f\|_\mathcal{H} \sqrt{\langle k(t, \cdot), k(t, \cdot) \rangle_\mathcal{H}}$$
$$= \|f\|_\mathcal{H} \sqrt{k(t, t)}$$
$$\le \|f\|_\mathcal{H} \sqrt{L_k \|t\|^\xi + C_k} \text{ by assumption}$$
$$\le C_0 + C_1 \|t\|^{|\xi|/2} \text{ for some } C_1, C_2 > 0.$$

We can write $T_Q = (T_{Q1}, \ldots, T_{QD})$. Let $e_d$ be a standard Euclidean basis vector. Then by the assumption and setting $u = e_d$ we have

$$P(|T_{Qd}| \geq z) \leq C_q \exp(-vz^\eta)$$

Letting $Q_d$ be the marginal law,

$$\int_S \exp(f(t))dQ \leq \int_S \exp(C_0 + C_1 \|t\|^{\xi/2})dQ$$

$$= \exp(C_0) \int_S \exp(C_1 \|t\|^{\xi/2})dQ$$

$$= \exp(C_0)\mathbb{E}\exp(C_1 \|T_Q\|^{\xi/2})$$

$$\leq \exp(C_0)\mathbb{E}\exp(C_1(\sqrt{d} \max_{d=1,\ldots,D} |T_{Qd}|)^{\xi/2})$$

$$\leq \exp(C_0)\sum_{d=1}^D \mathbb{E}\exp(C_2|T_{Qd}|^{\xi/2})$$

which will be finite if each $\mathbb{E}\exp(C_2|T_{Qd}|^{\xi/2}) < \infty$. Now letting $S_d$ be the relevant dimension of $S$,

$$\mathbb{E}\exp(C_2|T_{Qd}|^{\xi/2}) = \int_{S_d} \exp(C_2|t_d|^{\xi/2})dQ_d$$

$$\leq \sum_{j=-\infty}^{-1} \int_j^{j+1} \exp(C_2|t_d|^{\xi/2})dQ_d + \sum_{j=0}^{\infty} \int_j^{j+1} \exp(C_2|t_d|^{\xi/2})dQ_d$$

where the inequality follows since $S_d \subseteq \mathbb{R}$, $\exp$ is a non-negative function and probability measures are monotonic. We will show that the second sum converges. Similar techniques can be shown for the first sum. Note that for $j \geq 0$

$$Q_d([j, j+1)) = P(T_d \geq j) - P(T_d \geq j+1)$$

$$\leq P(T_d \geq j)$$

$$\leq C_q \exp(-vj^\eta) \text{ by assumption}$$

Then

$$\sum_{j=0}^{\infty} \int_j^{j+1} \exp(C_2|t_d|^{\xi/2})dQ_d \leq \sum_{j=0}^{\infty} \exp(C_2|j|^{\xi/2})Q_d([j, j+1))$$

$$\leq \sum_{j=0}^{\infty} C_Q \exp(C_2|j|^{\xi/2} - vj^\eta)$$

Let $a_j = \exp(C_2|j|^{\xi/2} - vi^\eta)$. We will use the ratio test to show that the RHS converges. We have

$$\left|\frac{a_{j+1}}{a_j}\right| = \exp(C_2((j+1)^{\xi/2} - j^{\xi/2}) - v[(j+1)^\eta - j^\eta]). \tag{11}$$

We want this ratio to be $< 1$ for large $j$. We thus need to select $\eta$ so that for sufficiently large $j$, we have

$$\frac{C_1}{v}((j+1)^{\xi/2} - j^{\xi/2}) < [(j+1)^\eta - j^\eta].$$

Assume that $\eta > \frac{\xi}{2}$. Then

$$\frac{(j+1)^\eta - j^\eta}{(j+1)^{\xi/2} - j^{\xi/2}} = \frac{j^\eta[(1+\frac{1}{j})^\eta - 1]}{j^{\xi/2}[(1+\frac{1}{j})^{\xi/2} - 1]}$$

$$\geq j^{\eta - \xi/2}.$$

Since the RHS is unbounded for $\eta > \frac{\xi}{2}$, we have that Eqn. 11 holds for sufficiently large $j$. By the ratio test $\mathbb{E}_{q_d(t)} \exp(C_2|T_d|^{\xi/2}) = \sum_{j=-\infty}^{-1} \int_j^{j+1} \exp(C_2|t_d|^{\xi/2}) dQ_d + \sum_{j=0}^{\infty} \int_j^{j+1} \exp(C_2|t_d|^{\xi/2}) dQ_d$ is finite. Thus putting everything together we have

$$
\int_S \exp(f(t)) dQ \leq \int_S \exp(C_0 + C_1 \|t\|^{\xi/2}) dQ
$$
$$
< \exp(C_0) \sum_{d=1}^{D} \mathbb{E} \exp(C_2|T_{Qd}|^{\xi/2})
$$
$$
< \infty
$$

and $\tilde{p}(t)$ can be normalized.

$\square$

## C.2 Proof of Corollary 5.2

Here we prove two special cases of kernel growth rate and tail decay.

*Proof.* Let $\xi = 4$. Then $\eta > 2$ and

$$
P(|u^T T| > t) \leq P(|u^T T| \geq t) \text{ monotonicity}
$$
$$
\leq C_Q \exp(-vt^\eta)
$$
$$
< C_Q \exp(-vt^2).
$$

The second case is similar. For the uniformly bounded kernel,

$$
\int_S \exp(\langle f, k(\cdot, t) \rangle_{\mathcal{H}}) dQ \leq \exp(\|f\|_{\mathcal{H}} \sqrt{C_k}) \int_S dQ
$$
$$
= \exp(\|f\|_{\mathcal{H}} \sqrt{C_k})
$$
$$
< \infty.
$$

The first line follows from Cauchy Schwarz and $\xi = 0$

$\square$

## D   Proofs Related to Kernel Deformed Exponential Family

### D.1   Proof of Lemma 5.3

Here we prove a key equality for calculating normalization of deformed exponential families.

*Proof.* The high level idea is to express a term inside the deformed exponential family that becomes $1/Z$ once outside. Recalling the definition of $\beta$-exponential in Eqn. 5,

$$
\exp_{2-\alpha}(f(t) - \log_\alpha(Z)) = [1 + (\alpha - 1)(f(t) - \log_\alpha Z)]_+^{\frac{1}{\alpha-1}}
$$
$$
= [1 + (\alpha - 1)f(t) - (\alpha - 1)\frac{Z^{1-\alpha} - 1}{1 - \alpha}]_+^{\frac{1}{\alpha-1}}
$$
$$
= [1 + (\alpha - 1)f(t) + Z^{1-\alpha} - 1]_+^{\frac{1}{\alpha-1}}
$$
$$
= [(\alpha - 1)f(t) + Z^{1-\alpha}]_+^{\frac{1}{\alpha-1}}
$$
$$
= [(\alpha - 1)f(t)\frac{Z^{\alpha-1}}{Z^{\alpha-1}} + Z^{1-\alpha}]_+^{\frac{1}{\alpha-1}}
$$
$$
= \frac{1}{Z}[(\alpha - 1)f(t)Z^{\alpha-1} + 1]_+^{\frac{1}{\alpha-1}}
$$
$$
= \frac{1}{Z} \exp_{2-\alpha}(Z^{\alpha-1}f(t)).
$$

Note that one could potentially also prove this starting with Proposition 1 and B.3 from [23]. However after some investigation we found our approach to be easier.

$\square$

## D.2 Proof of Proposition 5.4

Here we prove that if $\tilde{f}(t)$ has compact support and the kernel grows sufficiently slowly, then its deformed exponential has a finite integral.

*Proof.*

$$\int_S \exp_{2-\alpha}(\tilde{f}(t))dQ = \int_S [1 + (\alpha - 1)\tilde{f}(t)]_+^{\frac{1}{\alpha-1}} dQ$$

$$= \int_{\|t\| \leq C_t} [1 + (\alpha - 1)\tilde{f}(t)]_+^{\frac{1}{\alpha-1}} dQ$$

$$\leq \int_{\|t\| \leq C_t} [1 + (\alpha - 1)(C_0 + C_1|C_t|^{\xi/2})]_+^{\frac{1}{\alpha-1}} dQ$$

$$< \infty$$

where the 2nd to last line is due to the inequality $|f(t)| \leq C_0 + C_1\|t\|^{\xi/2}$ from the proof in Appendix C.1. $\qquad\square$

## D.3 Proof of Corollary 5.5

Here we prove that we can normalize deformed exponential families under the previous conditions.

*Proof.* From proposition 5.4 and the assumption,

$$\int_S \exp_{2-\alpha}(\tilde{f}(t))dQ = Z$$

for some $Z > 0$. Then

$$\int_S \frac{1}{Z} \exp_{2-\alpha}(Z^{\alpha-1}f(t))dQ = 1$$

$$\int_S \exp_{2-\alpha}(f(t) - \log_\alpha Z)dQ = 1$$

where the second line follows from lemma 5.3. Set $A_\alpha(f) = \log_\alpha(Z)$ and we are done. $\qquad\square$

## D.4 Approximation Theory

This section proves Theorem 5.6. We start with a Proposition, which says that under a kernel integration condition, deformed exponential families parametrized by functions in $\mathcal{H}$ can approximate similar densities parametrized by functions in $C_0(S)$ arbitrarily well.

**Proposition D.1.** *Define*

$$\mathcal{P}_0 = \{\pi_f(t) = \exp_{2-\alpha}(f(t) - A_\alpha(f)), t \in S : f \in C_0(S)\}$$

*where $S \subseteq \mathbb{R}^d$. Suppose $k(x, \cdot) \in C_0(S), \forall x \in S$ and*

$$\int \int k(x, y)d\mu(x)d\mu(y) > 0, \forall \mu \in M_b(S)\backslash\{0\}. \tag{12}$$

*here $M_b(S)$ is the space of bounded measures over $S$. Then the set of deformed exponential families is dense in $\mathcal{P}_0$ wrt $L^r(Q)$ norm and Hellinger distance.*

*Proof.* The kernel integration condition tells us that $\mathcal{H}$ is dense in $C_0(S)$ with respect to $L^\infty$ norm. This was shown in [34]. For the $L^r$ norm, we apply $\|p_f - p_g\|_{L^r} \leq 2M_{\exp}\|f - g\|_\infty$ from Lemma D.5 with $f \in C_0(S)$, $g \in \mathcal{H}$, and $f_0 = f$. $L^1$ convergence implies Hellinger convergence.

$\qquad\square$

We can then use this for our main proof of Theorem 5.6, which we restate here for reference. Note that our Bregman divergence result is analogous to [33]'s KL divergence result. KL divergence is Bregman divergence with the Shannon entropy functional: we show the same for Tsallis entropy, which is maximized given expected sufficient statistics by deformed exponential families [27]. The Bregman divergence describes how close a density's uncertainty is to its first order approximation evaluated at another density.

**Theorem D.2.** *Let $q_0 \in C(S)$, such that $q_0(t) > 0$ for all $t \in S$, where $S \subseteq \mathbb{R}^d$ is locally compact Hausdorff and $q_0$ is the Radon Nikodym derivative of measure $Q$ with respect to a dominating measure $\nu$. Suppose there exists $l > 0$ such that for any $\epsilon > 0, \exists R > 0$ satisfying $|p(t) - l| \leq \epsilon$ for any $t$ with $\|t\|_2 > R$. Define*

$$\mathcal{P}_c = \{p \in C(S) : \int_S p(t)dQ = 1, p(t) \geq 0, \forall t \in S \text{ and } p - l \in C_0(S)\}.$$

*Suppose $k(t, \cdot) \in C_0(S)\forall t \in S$ and the kernel integration condition (Eqn. 12) holds. Then kernel deformed exponential families are dense in $\mathcal{P}_c$ wrt $L^r$ norm, Hellinger distance and Bregman divergence for the $\alpha$-Tsallis negative entropy functional.*

*Proof.* For any $p \in \mathcal{P}_c$, define $p_\delta = \frac{p+\delta}{1+\delta}$. Then

$$\|p - p_\delta\|_r = \left\| p - \frac{p+\delta}{1+\delta} \right\|_r$$
$$= \left\| \frac{p(1+\delta)}{1+\delta} - \frac{p+\delta}{1+\delta} \right\|$$
$$= \frac{\delta}{1+\delta}\|p - 1\|_r$$
$$\to 0$$

for $1 \leq r \leq \infty$. Thus for any $\epsilon > 0, \exists \delta_\epsilon > 0$ such that for any $0 < \theta < \delta_\epsilon$, we have $\|p - p_\theta\|_r \leq \epsilon$, where $p_\theta(t) > 0$ for all $t \in S$.

Define $f = \left( \frac{1+\theta}{l+\theta} \right)^{1-\alpha} \log_{2-\alpha} p_\theta \frac{1+\theta}{l+\theta}$. Since $p \in C(S)$, so is $f$. Fix any $\eta > 0$ and note that

$$f(t) \geq \eta$$
$$\left( \frac{1+\theta}{l+\theta} \right)^{1-\alpha} \log_{2-\alpha} p_\theta \frac{1+\theta}{l+\theta} \geq \eta$$
$$\log_{2-\alpha} p_\theta \frac{1+\theta}{l+\theta} \geq \left( \frac{1+\theta}{l+\theta} \right)^{\alpha-1} \eta$$
$$p_\theta \frac{1+\theta}{l+\theta} \geq \exp_{2-\alpha}\left( \left( \frac{1+\theta}{l+\theta} \right)^{\alpha-1} \eta \right)$$
$$p_\theta \geq \frac{l+\theta}{1+\theta} \exp_{2-\alpha}\left( \left( \frac{1+\theta}{l+\theta} \right)^{\alpha-1} \eta \right)$$
$$p_\theta(1+\theta) \geq (l+\theta) \exp_{2-\alpha}\left( \left( \frac{1+\theta}{l+\theta} \right)^{\alpha-1} \eta \right)$$
$$\frac{p+\theta}{1+\theta}(1+\theta) \geq (l+\theta) \exp_{2-\alpha}\left( \left( \frac{1+\theta}{l+\theta} \right)^{\alpha-1} \eta \right)$$
$$p - l \geq (l+\theta)\left( \exp_{2-\alpha}\left( \left( \frac{1+\theta}{l+\theta} \right)^{\alpha-1} \eta \right) - 1 \right)$$

Thus

$$A = \{t : f(t) \geq \eta\}$$
$$= \left\{ p - l \geq (l+\theta)\left( \exp_{2-\alpha}\left( \left( \frac{1+\theta}{l+\theta} \right)^{\alpha-1} \eta \right) - 1 \right) \right\}$$

Since $p - l \in C_0(S)$ the set on the second line is bounded. Thus $A$ is bounded so that $f \in C_0(S)$. Further, by Lemma 5.3

$$p_\theta = \exp_{2-\alpha} \left( f - \log_\alpha \frac{1+\theta}{l+\theta} \right)$$

giving us $p_\theta \in \mathcal{P}_0$. By Proposition D.1 there is some $p_g$ in the deformed kernel exponential family so that $\|p_\theta - p_g\|_{L^r(S)} \le \epsilon$. Thus $\|p - p_g\|_r \le 2\epsilon$ for any $1 \le r \le \infty$. To show convergence in Helinger distance, note

$$\begin{aligned}
H^2(p, p_g) &= \frac{1}{2} \int_S (\sqrt{p} - \sqrt{p_g})^2 dQ \\
&= \frac{1}{2} \int_S (p - 2\sqrt{pp_g} + p_g)dQ \\
&\le \frac{1}{2} \int_S (p - 2\min(p, p_g) + p_g)dQ \\
&= \frac{1}{2} \int_S |p - p_g| dQ \\
&= \frac{1}{2} \|p - p_g\|_1
\end{aligned}$$

so that $L^1(S)$ convergence, which we showed, implies Hellinger convergence. Let us consider the Bregman divergence. Note the generalized triangle inequality[7] for Bregman divergence

$$B_{\Omega_\alpha}(p, p_g) = \underbrace{B_{\Omega_\alpha}(p, p_\theta)}_{I} + \underbrace{B_{\Omega_\alpha}(p_\theta, p_g)}_{II} - \underbrace{\langle p - p_\theta, \nabla\Omega_\alpha(p_\theta) - \nabla\Omega_\alpha(p_g)\rangle_2}_{III} \tag{13}$$

**Term I**

$$\begin{aligned}
B_{\Omega_\alpha}(p, p_\theta) &= \frac{1}{\alpha(\alpha-1)} \int_S (p^\alpha - p_\theta^\alpha)dQ - \langle \nabla\Omega_\alpha(p_\theta), p - p_\theta \rangle \\
&= \frac{1}{\alpha(\alpha-1)} \int_S (p^\alpha - p_\theta^\alpha)dQ - \frac{1}{\alpha-1} \int p_\theta^{\alpha-1}(p - p_\theta)dQ \\
&\le \frac{1}{\alpha(\alpha-1)} \int_S |p^\alpha - p_\theta^\alpha| dQ + \frac{1}{\alpha-1} \|p_\theta^{\alpha-1}\|_1 \|p - p_\theta\|_\infty
\end{aligned}$$

Note that the Bregman divergence is non-negative and thus we only need to worry about an upper bound. The first term on the rhs clearly vanishes as $\theta \to 0$ since $p_\theta \to p$ and we can pull the limit under the integral since $p, p_\theta$ are bounded. For the second term, we already showed that $\|p - p_\theta\|_\infty \to 0$ as $\theta \to 0$.

**Term II**

Fix $\theta$. Then term $II$ converges to 0 by Lemma D.5.

**Term III**

For term $III$,
$$\langle p - p_\theta, \nabla\Omega_\alpha(p_\theta) - \nabla\Omega_\alpha(p_g)\rangle_2 \le \|p - p_\theta\|_\infty \|\nabla\Omega_\alpha(p_\theta) - \nabla\Omega_\alpha(p_g)\|_1$$

Clearly the first term on the rhs converges by $L^r$ convergence. The $L^1$ term for the gradient is given by

$$\begin{aligned}
\|\nabla\Omega_\alpha(p_\theta) - \nabla\Omega_\alpha(p_g)\|_1 &= \frac{1}{\alpha-1} \int |p_\theta(t)^{\alpha-1} - p_g(t)^{\alpha-1}| dQ \\
&\le \int (\|p_\theta\|_\infty + \|p_\theta - p_g\|_\infty)^{\alpha-2} \|p_\theta - p_g\|_\infty dQ \quad \text{Eqn. 17} \\
&= (\|p_\theta\|_\infty + \|p_\theta - p_g\|_\infty)^{\alpha-2} \|p_\theta - p_g\|_\infty
\end{aligned}$$

so that the inner product terms are bounded as

$$|\langle p - p_\theta, \nabla\Omega_\alpha(p_\theta) - \nabla\Omega_\alpha(p_g)\rangle_2| \le (\|p_\theta\|_\infty + \|p_\theta - p_g\|_\infty)^{\alpha-2} \|p_\theta - p_g\|_\infty \|p - p_\theta\|_\infty$$

Fixing $\theta$ and letting $g \to \theta$ the RHS goes to 0. $\qquad\square$

---

[7]actually an equality, see https://www2.cs.uic.edu/ zhangx/teaching/bregman.pdf for proof

### D.4.1 Some Supporting Lemmas and Claims for Approximation Theory

**Lemma D.3.** *(Functional Mean Value Theorem) Let $F : X \to \mathbb{R}$ be a Gateaux differentiable functional where $f, g \in X$ some Banach space with norm $\| \cdot \|$. Then*

$$|F(f) - F(g)| \leq \|F'(h)\|_{op}\|f - g\|$$

*where $h = g + c(f - g)$ for some $c \in [0, 1]$, $F'(h)$ is the Gateaux derivative of $F$, and $\| \cdot \|_{op}$ is the operator norm $\|A\|_{op} = \inf\{c > 0 : \|Ax\| \leq c\|x\| \forall x \in X\}$.*

*Proof.* Consider $G(\eta) = F(g + \eta(f - g))$. Apply the ordinary mean value theorem to obtain

$$G(1) - G(0) = G'(c), c \in [0, 1]$$
$$= F'(g + c(f - g)) \cdot (f - g)$$

and thus

$$|F(f) - F(g)| \leq \|F'(h)\|_{op}\|f - g\|$$

$\square$

*Claim* 1. Consider $\mathcal{P}_\infty = \{p_f = \exp_{2-\alpha}(f - A_\alpha(f)) : f \in L^\infty(S)\}$. Then for $p_f \in \mathcal{P}_\infty$, $A_\alpha(f) \leq \|f\|_\infty$.

*Proof.*

$$p_f(t) = \exp_{2-\alpha}(f(t) - A_\alpha(f))$$
$$\leq \exp_{2-\alpha}(\|f\|_\infty - A_\alpha(f)) \text{ for } 1 < \alpha \leq 2$$
$$\int_S p_f(t)dQ \leq \int_S \exp_{2-\alpha}(\|f\|_\infty - A_\alpha(f))dQ$$
$$1 \leq \exp_{2-\alpha}(\|f\|_\infty - A_\alpha(f))$$
$$\log_{2-\alpha} 1 \leq \|f\|_\infty - A_\alpha(f)$$
$$A_\alpha(f) \leq \|f\|_\infty$$

where for the second line recall that we assumed that throughout the paper $1 < \alpha \leq 2$. $\square$

**Lemma D.4.** *Consider $\mathcal{P}_\infty = \{p_f = \exp_{2-\alpha}(f - A_\alpha(f)) : f \in L^\infty(S)\}$. Then the Gateaux derivative of $A_\alpha : L^\infty \to \mathbb{R}$ is given by the map*

$$A'(f)(g) = \mathbb{E}_{\tilde{p}_f^{2-\alpha}}(g(T))$$
$$= \frac{\int p_f^{2-\alpha}(t)g(t)dQ}{\int p_f^{2-\alpha}(t)dQ}$$

*Proof.* This proof has several parts. We first derive the Gateaux differential of $p_f$ in a direction $\psi \in L^\infty$ and as it depends on the Gateaux differential of $A_\alpha(f)$ in that direction, we can rearrange terms to recover the latter. We then show that it exists for any $f, \psi \in L^\infty$. Next we show that the second Gateaux differential of $A_\alpha(f)$ exists, and use that along with a functional Taylor expansion to prove that the first Gateaux derivative is in fact a Frechet derivative.

In [23] they show how to compute the gradient of $A_\alpha(\theta)$ for the finite dimensional case: we extend this to the Gateaux differential. We start by computing the Gateaux differential of $p_f$.

$$\frac{d}{d\eta}p_{f+\eta\psi}(t) = \frac{d}{d\eta}\exp_{2-\alpha}(f(t) + \eta\psi(t) - A_\alpha(f + \eta\psi))$$
$$= \frac{d}{d\eta}[1 + (\alpha - 1)(f(t) + \eta\psi(t) - A_\alpha(f + \eta\psi))]_+^{1/(\alpha-1)}$$
$$= [1 + (\alpha - 1)(f(t) + \eta\psi(t) - A_\alpha(f + \eta\psi))]_+^{(2-\alpha)/(\alpha-1)} \left(\psi(t) - \frac{d}{d\eta}A_\alpha(f + \eta\psi)\right)$$
$$= p_{f+\eta\psi}^{2-\alpha}(t) \left(\psi(t) - \frac{d}{d\eta}A_\alpha(f + \eta\psi)\right)$$

evaluating at $\eta = 0$ gives us

$$dp(f; \psi)(t) = p_f^{2-\alpha}(t)\,(\psi(t) + dA_\alpha(f; \psi))$$

Note that since $p_{f+\eta\psi}(t)$ is a probability density function, it is integrable and thus we can apply the dominated convergence theorem to pull a derivative with respect to $\eta$ under an integral. Noting that the integral of a density function is 1 and thus its derivative is 0, we can then recover the Gateaux diferential of $A_\alpha$ via

$$0 = \frac{d}{d\eta}\bigg|_{\eta=0} \int p_{f+\eta\psi}(t)dQ$$

$$= \int dp(f; \psi)(t)dQ$$

$$= \int p_f(t)^{2-\alpha}(\psi(t) - dA_\alpha(f; \psi))dQ$$

$$dA_\alpha(f; \psi) = \mathbb{E}_{\tilde{p}_f^{2-\alpha}}(\psi(T))$$

$$< \infty$$

where the last line follows as $\psi \in L^\infty$. Thus the Gateaux derivative exists in $L^\infty$ directions. The derivative at $f$ maps $\psi :\to \mathbb{E}_{\tilde{p}_f^{2-\alpha}}(\psi(T))$ i.e. $A'_\alpha(f)(\psi) = \mathbb{E}_{\tilde{p}_f^{2-\alpha}}(\psi(T))$.

$\square$

**Lemma D.5.** *Define $\mathcal{P}_\infty = \{p_f = \exp_{2-\alpha}(f - A_\alpha(f)) : f \in L^\infty(S)\}$ where $L^\infty(S)$ is the space of almost everywhere bounded measurable functions with domain $S$. Fix $f_0 \in L^\infty$. Then for any fixed $\epsilon > 0$ and $p_g, p_f \in \mathcal{P}_\infty$ such that $f, g \in \overline{B}_\epsilon^\infty(f_0)$ the $L^\infty$ closed ball around $f_0$, there exists constant $M_{\exp} > 0$ depending only on $f_0$ such that*

$$\|p_f - p_g\|_{L^r} \leq 2M_{\exp}\|f - g\|_\infty$$

*Further*

$$B_{\Omega_\alpha}(p_f, p_g) \leq \frac{1}{\alpha - 1}\|p_f - p_g\|_\infty[(\|p_f\|_\infty + \|p_f - p_g\|_\infty)^{\alpha-1} + \exp_{2-\alpha}(2\|g\|_\infty)]$$

*Proof.* This Lemma mirrors Lemma A.1 in [33], but the proof is very different as they rely on the property that $\exp(x + y) = \exp(x)\exp(y)$, which does not hold for $\beta$-exponentials. We thus had to strengthen the assumption to include that $f$ and $g$ lie in a closed ball, and then use the functional mean value theorem Lemma D.3 as the main technique to achieve our result.

Consider that by the mean value inequality,

$$|p_f(t) - p_g(t)| = |\exp_\beta(f(t) - A_\alpha(f)) - \exp_\beta(g(t) - A_\alpha(g))|$$

$$\leq |\exp_\beta(h(t) - A_\alpha(h))^{2-\alpha}||f(t) - A_\alpha(f) - (g(t) - A_\alpha(g))|$$

$$\leq \|\exp_\beta(h - A_\alpha(h))^{2-\alpha}\|_\infty(\|f - g\|_\infty + |A_\alpha(f) - A_\alpha(g)|)$$

where $h = cf + (1 - c)g$ for some $c \in [0, 1]$. This implies

$$\|p_f - p_g\|_{L^r} = \|\exp_\beta(f - A_\alpha(f)) - \exp_\beta(g - A_\alpha(g))\|_{L^r}$$

$$\leq \|\exp_\beta(h - A_\alpha(h))^{2-\alpha}\|_\infty(\|f - g\|_\infty + |A_\alpha(f) - A_\alpha(g)|) \qquad (14)$$

We need to bound $\exp_\beta(h - A_\alpha(h))$ and $\|A_\alpha(f) - A_\alpha(g)\|_\infty$.

We can show a bound on $\|h\|_\infty$

$$\|h\|_\infty = \|cf + (1 - c)g - f_0 + f_0\|_\infty$$

$$\leq \|c(f - f_0) + (1 - c)(g - f_0) + f_0\|_\infty$$

$$\leq c\|f - f_0\|_\infty + (1 - c)\|g - f_0\|_\infty + \|f_0\|_\infty$$

$$\leq \epsilon + \|f_0\|_\infty$$

so that $h$ is bounded. Now we previously showed in claim 1 that $|A_\alpha(h)| \leq \|h\|_\infty \leq \epsilon + \|f_0\|_\infty$. Since $h, A_\alpha(h)$ are both bounded $\exp_\beta(h - A_\alpha(h))^{2-\alpha}$ is also.

Now note that by Lemma D.3,
$$|A_\alpha(f) - A_\alpha(g)| \le \|A'_\alpha(h)\|_{\text{op}}\|f - g\|_\infty$$

We need to show that $\|A'_\alpha(h)\|_{\text{op}}$ is bounded for $f, g \in \overline{B}_\epsilon(f_0)$. Note that in Lemma D.4 we showed that

$$|A'_\alpha(f)(g)| = |\mathbb{E}_{p_f^{2-\alpha}}[g(T)]|$$
$$\le \|g\|_\infty$$

Thus $\|A'_\alpha\|_{\text{op}} = \sup\{|A'_\alpha(h)(m)| : \|m\|_\infty = 1\} \le 1$. Let $M_{\exp}$ be the bound on $\exp_\beta(h - A_\alpha(h))$. Then putting everything together we have the desired result

$$\|p_f - p_g\|_{L^r} \le 2M_{\exp}\|f - g\|_\infty$$

Now

$$B_{\Omega_\alpha}(p_f, p_g) = \Omega_\alpha(p_f) - \Omega_\alpha(p_g) - \langle \nabla\Omega_\alpha(p_g), p_f - p_g \rangle_2 \tag{15}$$

For the inner prodct term, first note that following [23] the gradient is given by

$$\nabla\Omega_\alpha(p_g)(t) = \frac{p_g(t)^{\alpha-1}}{\alpha - 1} \tag{16}$$

Thus

$$|\langle \nabla\Omega_\alpha(p_g), p_f - p_g \rangle_2| \le \|\nabla\Omega_\alpha(p_g)\|_1\|p_f - p_g\|_\infty$$
$$= \frac{1}{\alpha - 1}\int_S \exp_{2-\alpha}(g(t) - A(g))dQ\|p_f - p_g\|_\infty$$
$$\le \frac{1}{\alpha - 1}\exp_{2-\alpha}(2\|g\|_\infty)\|p_f - p_g\|_\infty$$

where the second line follows from claim 1. Further note that by Taylor's theorem,

$$y^\alpha = x^\alpha + \alpha z^{\alpha-1}(y - x)$$

for some $z$ between $x$ and $y$. Then letting $y = p_f(t)$ and $x = p_g(t)$, we have for some $z = h(t)$ lying between $p_f(t)$ and $p_g(t)$ that

$$p_f(t)^\alpha = p_g(t)^\alpha + \alpha h(t)^{\alpha-1}(p_f(t) - p_g(t))$$

Since $f \in L^\infty$ then applying Claim 1 we have that each $p_f, p_g \in L^\infty$ and thus $h$ is. Then

$$|p_f(t)^\alpha - p_g(t)^\alpha| = \alpha|h(t)|^{\alpha-1}|p_f(t) - p_g(t)|$$
$$\le \alpha\|h\|_\infty^{\alpha-1}\|p_f - p_g\|_\infty$$
$$\le \alpha\max\{\|p_f\|_\infty, \|p_g\|_\infty\}^{\alpha-1}\|p_f - p_g\|_\infty$$
$$\le \alpha(\|p_f\|_\infty + \|p_f - p_g\|_\infty)^{\alpha-1}\|p_f - p_g\|_\infty \tag{17}$$

so that

$$|\Omega_\alpha(p_f) - \Omega_\alpha(p_g)| = \left|\frac{1}{\alpha(\alpha-1)}\int (p_f(t)^\alpha - p_g(t)^\alpha)dQ\right|$$
$$\le \frac{1}{\alpha - 1}(\|p_f\|_\infty + \|p_f - p_g\|_\infty)^{\alpha-1}\|p_f - p_g\|_\infty.$$

Putting it all together we obtain

$$B_{\Omega_\alpha}(p_f, p_g) \le \frac{1}{\alpha - 1}(\|p_f\|_\infty + \|p_f - p_g\|_\infty)^{\alpha-1}\|p_f - p_g\|_\infty$$
$$+ \frac{1}{\alpha - 1}\exp_{2-\alpha}(2\|g\|_\infty)\|p_f - p_g\|_\infty$$
$$= \frac{1}{\alpha - 1}\|p_f - p_g\|_\infty[(\|p_f\|_\infty + \|p_f - p_g\|_\infty)^{\alpha-1} + \exp_{2-\alpha}(2\|g\|_\infty)]$$

$$\square$$

# E   Numerical Integration Stability and Convergence Analysis

## E.1   Stable Numerical Integration

One issue is numerical underflow when computing the normalizing constant. In the kernel exponential family case, if $|f(t)|$ is very large and $f(t) < 0$ for all evaluated $t$, then on a computer $\exp(f(t))$ will round to $0$ for all the evaluated $t$. Thus $Z$ the normalizing constant will be estimated as $0$ in numerical integration and the estimate of $\exp(f(t))/Z$ cannot be computed. A similar issue exists for the deformed case.

For kernel exponential family, we can use the standard technique used in discrete softmax implementations (see [10] chapter 4). Note

$$
\begin{aligned}
\exp(f(t) - A(f)) &= \exp(f(t) - C + C - A(f)) \\
&= \exp(f(t) - C)\exp(C - A(f))
\end{aligned}
$$

Then instead of normalizing $\exp(f(t))$ with $\exp(-A(f))$ we normalize $\exp(f(t) - C)$ with $\exp(C - A(f))$, letting $C = \sup_t f(t)$. Taking $C = \sup_t f(t)$ will prevent underflow as if $f(t) < 0 \forall t$, $f(t) - \sup f(t) \approx 0$ for some $t$ (equality if there is a maximum, which there always is when using a finite set of $t$ for numerical integration).

For the deformed case, recall that by Lemma 5.3, $\exp_{2-\alpha}(f(t) - \log_\alpha Z) = \frac{1}{Z}\exp_{2-\alpha}(Z^{\alpha-1}f(t))$ for $Z > 0$. Then letting $\log_\alpha C = \sup_t f(t)$ and noting that for $x, y > 0$, $\log_\alpha x - \log_\alpha y = \log_\alpha(x \oslash_\alpha y) = [x^{1-\alpha} - y^{1-\alpha} + 1]_+^{\frac{1}{1-\alpha}}$ [37],

$$
\begin{aligned}
\exp_{2-\alpha}(f(t) - \log_\alpha Z) &= \exp_{2-\alpha}(f(t) - \log_\alpha C + \log_\alpha C - \log_\alpha Z) \\
&= \exp_{2-\alpha}(f(t) - \log_\alpha C - (\log_\alpha Z - \log_\alpha C)) \\
&= \exp_{2-\alpha}(f(t) - \log_\alpha C - \log_\alpha (Z \oslash_\alpha C)) \\
&= \frac{1}{Z \oslash_\alpha C}\exp_{2-\alpha}\left(\left(\frac{1}{Z \oslash_\alpha C}\right)^{\alpha-1}(f(t) - \log_\alpha C)\right)
\end{aligned}
$$

Now consider $\tilde{f}(t) = \left(\frac{1}{Z \oslash_\alpha C}\right)^{\alpha-1} f(t)$. Then

$$
\sup_t \tilde{f}(t) = \left(\frac{1}{Z \oslash_\alpha C}\right)^{\alpha-1}\log_\alpha C
$$

We can thus estimate $\tilde{f}(t)$, subtract $\sup_t \tilde{f}(t)$ (max in practice), take the deformed exponential, and normalize. The computational operations are nearly identical to the kernel exponential family case, although the reasons for their validity are very different.

## E.2   Kernel Exponential Family Attention

Here we show conditions for which numerical integration of

$$
\int_S \exp(f(t))V(t)dQ = \int_{-\infty}^\infty \exp(f(t))V(t)q_0(t)dt
$$

using the trapezoidal rule is exponentially convergent. We start by restating a theorem. This says that the trapezoidal rule for numerical integration of holomorphic functions of sufficiently fast decay has exponential convergence. A version of this theorem comes from [39], but there are slight issues with the notation and conditions. A slightly revised statement is in the course notes of [14], which we follow here.

**Theorem E.1.** *Let $w : \mathbb{C} \to \mathbb{C}$ be analytic in the strip $S_b = \{z \in \mathbb{C} : |Im(z)| < a\}$ for some $a > 0$. Suppose further that $w(z) \to 0$ as $|z| \to \infty$ in the strip, and for some $M > 0$,*

$$
\int_{-\infty}^\infty |w(x + ib)|dx \le M \tag{18}
$$

*for all $b \in (-a, a)$. Then leting $I = \int_{-\infty}^\infty w(x)dx$ and $I_h = h\sum_{k=-\infty}^\infty w(kh)$,*

$$
|I_h - I| \le \frac{2M}{\exp(2\pi a/h) - 1}
$$

Next we give conditions on $q_0(t)$, $V(t)$, and $k$ so that numerical integration is exponentially convergent. Specifically that $q_0$ and $V$ have holomorphic complex extensions to a strip and that $k$ has exponentially decaying Fourier transform.

**Corollary E.2.** *Assume that $q_0, V$ have holomorphic complex extensions to a strip $S_b = \{z \in \mathbb{C} : |Im(z)| < b\}$, $q_0(z) \to 0$ as $|z| \to \infty$ and $V$ is bounded by $M_V$. Also assume that for any $t_i \in \mathbb{R}$ the absolute value of the Fourier transform $|\hat{k}(\xi, t_i)| = |\int_{-\infty}^{\infty} k(t, t_i) \exp(-2\pi i \xi t) dt| \leq |A(t_i) \exp(-a|\xi|)|$, where $A : \mathbb{R} \to \mathbb{C}$ and $a > 0$ is a fixed constant. Further assume that $f(t) = \sum_{i=1}^{I} \gamma_i k(t, , t_i)$, $k(t, t_i) \in C_0$, and*

$$\int_{-\infty}^{\infty} |q(x + ib)| dx \leq M \tag{19}$$

*for some $M > 0$. Then if $w(z) = q_0(z) \exp(f(z)) V(z) \to 0$ as $|z| \to \infty$, the trapezoidal rule for $w(t) = q_0(t) \exp(f(t)) V(t)$ is exponentially convergent.*

*Proof.* Note that

$$\hat{f}(\xi) = \int_{-\infty}^{\infty} f(t) \exp(-2\pi i \xi t) dt$$

$$= \int_{-\infty}^{\infty} \sum_{i=1}^{I} \gamma_i k(t, t_i) \exp(-2\pi i \xi t) dt$$

$$= \sum_{i=1}^{I} \gamma_i \int_{-\infty}^{\infty} k(t, t_i) \exp(-2\pi i \xi t) dt$$

$$= \sum_{i=1}^{I} \gamma_i \hat{k}(\xi, t_i)$$

and thus

$$|\hat{f}(\xi)| \leq \sum_{i=1}^{I} |\gamma_i| |A(t_i)| \exp(-a|\xi|)$$

$$\leq \exp(-a|\xi|) I \max_{i \in I} |\gamma_i| |A(t_i)|$$

Thus since the fourier transform of $f$ has (at least) exponential decay, by Theorem 3.1 in [35] the extension of $f(t)$ to $f(z)$ is holomorphic in the strip. Since compositions and products of holomorphic functions are holomorphic, $q_0(z) \exp(f(z)) V(z)$ is holomorphic in the strip. Further, since for each $t_i$, $k(t, t_i) \in C_0$, $f$ is and thus it is bounded and thus its complex extension is bounded, say by $M_F$, thus the complex $\exp(f)$ is bounded by $\exp(M_f)$. Since $V$ is also bounded, we have

$$\int_{-\infty}^{\infty} |w(x + ib)| dx \leq M_V \exp(M_f) M.$$

Finally,

$$|w(z)| \leq M_V \exp(M_f) |q(z)|$$
$$\to 0 \text{ as } |z| \to \infty$$

and thus $w$ satisfies the conditions of the previous theorem. □

Finally, we show example special cases of $q_0, V, k$ satisfying those conditions.

**Corollary E.3.** *If $k$ is a Gaussian kernel, $V$ is a linear combination of Gaussian RBFs, and $q_0(t) = \frac{1}{\sqrt{2\pi}} \exp(-t^2/2)$, then the trapezoidal rule for kernel exponential family attention is exponentially convergent.*

*Proof.* For the Gaussian kernel, we have $k(t, t_i) = \exp(-\zeta^2(t - t_i)^2)$. Now

$$\int_{-\infty}^{\infty} \exp(-\zeta^2(t - t_i)^2) \exp(-2\pi i \xi t) dt = \frac{\exp(-i\xi t_i)}{\gamma} \sqrt{\pi} \exp(-\xi^2/(4\zeta^2))$$

and thus since $\exp(-i\xi t_i)$ lies on the unit circle,

$$\left| \int_{-\infty}^{\infty} \exp(-\zeta^2(t - t_i)^2) \exp(-2\pi i \xi t) dt \right| \leq \frac{\sqrt{\pi}}{\gamma} \exp(-\xi^2/(4\zeta^2)).$$

This satisfies the Fourier decay condition for the kernel so that functions $f \in \mathcal{H}$ have holomorphic extensions to the strip. A similar technique can be applied to $V$ and $q_0$ to show that they have holomorphic extensions to the strip, and clearly both $V$ and $f$ are bounded since they are continuous and vanish at infinity. Thus $\exp(f)$ is also bounded. For the decay of $q_0$, note

$$\begin{aligned}
\frac{1}{\sqrt{2\pi}} \int_{-\infty}^{\infty} |\exp(-(x + ib)^2/2)| dx &= \frac{1}{\sqrt{2\pi}} \int_{-\infty}^{\infty} |\exp((-x^2 + b^2 - 2xbi)/2)| dx \\
&\leq \frac{1}{\sqrt{2\pi}} \exp(b^2/2) \int_{-\infty}^{\infty} \exp(-x^2/2) dx \\
&= \frac{1}{\sqrt{2\pi}} \exp(b^2/2) \\
&\leq \frac{1}{\sqrt{2\pi}} \exp(a^2/2)
\end{aligned}$$

satisfying the integration condition Eqn. 18. It remains to show that $w(z) \to 0$. First note that

$$\begin{aligned}
|q_0(z)| &= \frac{1}{\sqrt{2\pi}} |\exp(-(x + ib)^2/2)| \\
&\leq \frac{1}{\sqrt{2\pi}} \exp(a^2/2) \exp(-x^2/2)
\end{aligned}$$

Since $z \in S_b$, $|z| \to \infty$ requires $|x| \to \infty$, and $\frac{1}{\sqrt{2\pi}} \exp(a^2/2) \exp(-x^2/2) \to 0$ as $|x| \to \infty$. Thus $q_0(z) \to 0$ as $|z| \to \infty$ in the strip. Note that since $V, f$ are bounded and $q_0(z) \to 0$ in the strip as $|z| \to \infty$, $w(z) \to 0$ as $|z| \to \infty$ in the strip.

### E.3 Convergence of Numerical Integration for Kernel Deformed Exponential Family Attention

#### E.3.1 Smooth Approximation to Kernel Deformed Exponential Family Attention

**Definition E.4.** Let

$$\exp_{\rho,\beta}(t) \equiv \left[ \frac{1}{\rho} \log(1 + \exp(\rho[1 + (1 - \beta)t])) \right]^{\frac{1}{1-\beta}} \tag{20}$$

for some $\rho > 0$.

*Claim 2.* $\exp_{\rho,\beta}(z)$ is holomorphic on the strip $\{z \in \mathbb{C} : \text{Im}(z) \in [-\pi/2, \pi/2]\}$ for the principal branch of $\log$.

*Proof.* This proof is adapted from [11]. We show it for $\exp_{0,0}$ but the idea can easily be extended to more general $\rho$ and $\beta$. Note that the principal branch of $\log z$ is analytic outside of $B = \{z = x + iy : -\infty < x \leq 0, y = 0\}$. Thus $\exp_{0,0}$ is analytic as long as $1 + \exp(1 + z)$ is not in $B$.

Now $\exp(1 + z) = \exp(x + 1) \exp(iy)$, so $\exp(1 + z)$ will be in $\{z = x + iy : x \in (-\infty, -1], y = 0\}$ when $\exp(x + 1) \geq 1$, $\exp(iy) = -1$, i.e. $x \geq -1$, $y = \pi + 2\pi n$, $n \in \mathbb{N}$. The $y$ condition will not be satisfied on the strip above and thus $\exp_{0,0}(z)$ is holomorphic on that strip. $\square$

**Corollary E.5.** *If $k$ is a Gaussian kernel, $V$ is a linear combination of Gaussian RBFs, and $q_0(t) = \frac{1}{\sqrt{2\pi}} \exp(-t^2/2)$, then the trapezoidal rule for softplus approximation to kernel deformed exponential family attention is exponentially convergent.*

*Proof.* Apply Theorem E.1. We already showed in the kernel exponential family section that $q_0$ and $V$ are holomorphic in the strip, and we showed that $\exp_{\rho,\beta}$ is in the previous claim. We also showed that $V$ is bounded and that $q_0$ satisfies the integration condition and $q_0(z) \to 0$ as $|z| \to \infty$ in the

strip. It remains to show that $\exp_{\rho,\beta}(f(t))$ is bounded (and thus its complex extension is). We show it for $\exp_{\rho,0}(f(t))$ but again the idea can be extended to general $\beta$. Note that since $\log(1+x) \leq x$ for $x > -1$,

$$\log(1 + \exp(1 + f(t))) \leq \exp(1 + f(t))$$
$$\leq \exp(1 + M_f)$$

since $f$ is bounded. Further, fixing $t$, $\log(1 + \exp(\rho(1+t)))/\rho$ is monotonically decreasing as a function of $\rho$ and thus for all $\rho \geq 1$,

$$\log(1 + \exp(\rho(1+t)))/\rho \leq \exp(1 + M_f)$$

Thus $w$ satisfies both the integration condition and the convergence condition. $\qquad\square$

$\square$

### E.3.2 Using the Smooth Approximation to Bound the Numerical Integral for Kernel Sparsemax Attention

Let $I_h$ be the numerical integral proportional to kernel deformed exponential family attention, $I_{h,s}$ be its softplus approximation, $I$ be the true kernel deformed exponential family integral, and $I_s$ be its softplus approximation. Then

$$|I_h - I| \leq |I_h - I_{h,s}| + |I_{h,s} - I_s| + |I_s - I|$$

We already bounded $|I_{h,s} - I_s|$ in the previous subsection, so we will bound the other two terms on the right hand side.

### E.3.3 Bounding $I_s - I$

We first bound the difference between the softplus integral and the integral using the positive part/ReLU. By Hoelder's inequality,

$$|\int_{-\infty}^{\infty} q_0(t) V(t)[\ln(1 + \exp(\rho(1+t)))/\rho - \max(0, 1+t)]dt|$$

$$\leq \|q_0 V\|_1 \text{esssup}_{t \in (-\infty,\infty)}[\ln(1 + \exp(\rho(1+t)))/\rho - \max(0, 1+t)]$$

$$\leq \|q_0 V\|_1 \ln(2)/\rho$$

### E.3.4 Bounding $I_h - I_{h,s}$

$$|I_{h,s} - I_h| = |h \sum_{k=-\infty}^{\infty} q_0(kh)V(kh)[\ln(1 + \exp(\rho(1+kh)))/\rho - \max(0, 1+kh)]|$$

$$= |h\langle q_0(h\cdot)V(h\cdot), \ln(1 + \exp(\rho(1 + h\cdot)))/\rho - \max(0, 1 + h\cdot)\rangle_{l^2}|$$

$$\leq h \left( \sum_{k=-\infty}^{\infty} q_0(hk)|V(hk)| \right) \ln(2)/\rho$$

where the last line again uses Hoelder's inequality.

### E.3.5 Putting it All Together

We now have, for any $\rho > 0$,

$$|I_h - I| \leq |I_h - I_{h,s}| + |I_{h,s} - I_s| + |I_s - I|$$

$$\leq \|q_0 V\|_1 \ln(2)/\rho + \frac{2M}{\exp(2\pi a/h) - 1} + h \left( \sum_{k=-\infty}^{\infty} q_0(hk)|V(hk)| \right) \ln(2)/\rho$$

and since this holds for all $\rho > 0$ we can take $\rho \to \infty$ and we have

$$|I_h - I| \leq \frac{2M}{\exp(2\pi a/h) - 1}$$

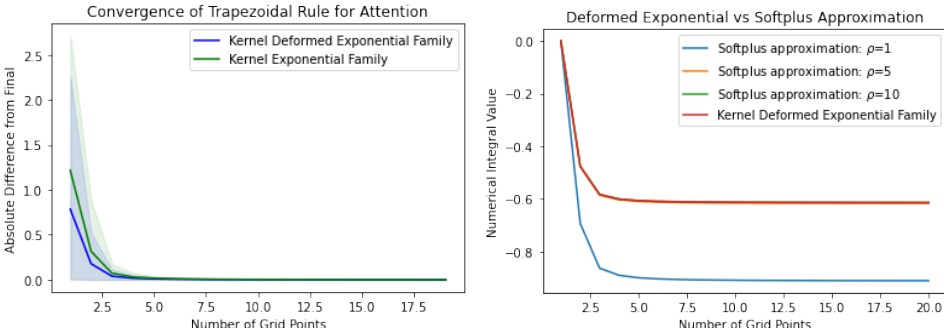

Figure 2: a) Convergence of trapezoidal rule for multimodal continuous attention. We see that both methods have very fast convergence empirically. The y-axis is the absolute difference between the numerical integral at $x$ vs the max value of $x$ ($x$ ranges from 1 to 19). The kernel deformed exponential family tends to converge faster than kernel exponential family attention. b) The value of the integral using kernel deformed exponential family sparsemax vs softplus approximations with different values of $\rho$. We see that the attention using softplus approximation becomes indistinguishable from the deformed exponential attention for $\rho = 5, 10$.

### E.4    Synthetic Experiments: Convergence

We now analyze convergence of the trapezoidal rule for multimodal continuous attention using numerical integration empirically using synthetic experiments. We define

$$f(t) = \sum_{i=1}^{I} \gamma_i k(t, t_i)$$

and

$$V(t) = \sum_{i=1}^{I} B_i k(t, t_i)$$

with $\gamma_i, B_i \sim U(-1, 1)$. We set $I = 10$ and $t_i$ to be evenly spaced in the interval $[0, 1]$. We then compute

$$\int_0^1 \exp(-t^2/2) V(t) \exp(f(t)) dt$$

which is proportional to attention for kernel exponential families. We also compute

$$\int_0^1 \exp(-t^2/2) V(t) \exp_0(f(t)) dt$$

proportional to the sparsemax case of kernel deformed exponential family attention. Finally, we also use the softplus approximation

$$\int_0^1 \exp(-t^2/2) V(t) \exp_{\rho,0}(f(t)) dt.$$

Note that we do not use the $\tilde{f}$ and $\tilde{\gamma}$ notation for the kernel deformed case as the main point is to examine the numerical integration convergence rather than to formally handle normalization. We simulate 100 times. Figure 2a compares the convergence as the number of grid points in the trapezoidal rule increases. 2b shows a single case of the integral value using deformed exponential vs the softplus approximation for various values of $\rho$ and numbers of grid points. We see that for $\rho = 5, 10$ the integral using softplus approximation is essentially indistinguishable from that using the positive part/ReLU.

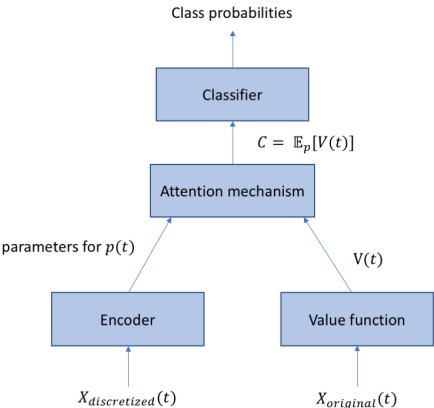

Figure 3: General architecture for classification using continuous attention mechanisms. The pipeline is trained end-to-end. The encoder takes a discretized representation of an observation (i.e. a time series itself, hidden units of an LSTM, or a layer of a CNN) and outputs parameters for an attention density. The value function takes the original (potentially irregularly sampled) time series or some representation and outputs parameters for a function $V(t)$. These are then combined in an attention mechanism by computing a context vector $c = \mathbb{E}_p[V(T)]$. For some parametrizations of $p$ and $V(t)$ this can be computed in closed form, while for others it must be done via numerical integration. The context vector is then fed into a classifier.

## F  Computational/Memory Complexity

Our attention mechanisms require computation in three steps:

1. Estimating the value function parameters with regularized multivariate linear regression. This was already required by [23, 24]. This involves computing $HF^T(FF^T + \lambda I_N)^{-1}$ for $F \in \mathbb{R}^{N \times L}, H \in \mathbb{R}^{D \times L}$. We thus need to compute

   - The Gram matrix $FF^T \in \mathbb{R}^{N \times N}$. This is $O(N^2 L)$.
   - The inverse $(FF^T + \lambda I_N)^{-1} \in \mathbb{R}^{N \times N}$. This is $O(N^3)$.
   - The matrix multiplication $HF^T \in \mathbb{R}^{D \times N}$. This is $O(DNL)$.
   - The matrix multiplication giving $HF^T(FF^T + \lambda I_N)^{-1} \in \mathbb{R}^{D \times N}$. This is $O(DNL)$.

   This gives a final cost of $O(N^3 + N^2 L + DNL)$. Note that if we do this for raw data, we only need to do this once, while if we do this for activations, we need to do it at every iteration.

2. Computing the normalizing constant of the attention density.

   - This requires computing $\exp(\tilde{f}(t))q_0(t)$ for each numerical integration grid point $1, \ldots, G$.
   - Each term has cost $O(I)$, where $I$ is the number of inducing points.

   The final cost is $O(GI)$.

3. Computing the numerical integral/context vector.

   - We need to compute $\exp(\tilde{f}(t))q_0(t)\psi_n(t)$ for each basis function $n = 1, \ldots, N$ and each numerical integration grid point $1, \ldots, G$.
   - Each such computation has cost $O(DI)$, since we have $D$ dimensions and $I$ inducing points.

   The total cost is thus $O(NGDI)$, linear in the number of basis functions, grid evaluation points, dimensions, and inducing points.

4. Applying the final linear transformation.

This gives total computation $O(N^3 + N^2 L + DNL + NGDI)$. The first three terms were already required in [23, 24], while the last is due to numerical integration. For memory, we require

| Type | Martins et al. | Ours |
|---|---|---|
| Computation | $O(N^3 + N^2L + DNL)$ | $O(N^3 + N^2L + DNL + NGDI)$ |
| Memory | $O(N^2 + ND)$ | $O(N^2 + NGD)$ |

Table 5: Computation and Memory requirements for our model vs [23]. $N$: basis functions $D$: dim $L$: sequence length $I$: inducing points $G$: integration grid points

.

| Experiment | Kernel Softmax | Kernel Sparsemax |
|---|---|---|
| MIT BIH | 25.70 | 27.20 |
| FordA | 0.70 | 0.68 |
| uWave | 1.26 | 1.25 |

Table 6: Wall clock time in seconds for one epoch of kernel softmax and kernel sparsemax on an A40. We do not display IMDB results due to having difficulties running it on recent versions of Pytorch/Python/Torchtext.

1. All the matrices of the value function. This gives $O(N^2 + DN)$.

2. All of the evaluations for numerical integration before integration happens. This requires $O(NGD)$. We are uncertain how these are implemented on a computer, and one may not need to store all the grid point evaluations at once for instance. Thus we can consider this to be an upper bound. Regardless neither memory requirement is excessive for most applications.

The total memory cost is thus $\boldsymbol{O(N^2 + NGD)}$, where the first term was already present in [23, 25] and the second is due to numerical integration. These results are summarized in Table 5.

## G   FordA: Additional Details

We plot attention densities for randomly selected examples under each density class. We find that only the kernel methods appear truly multimodal visually (Gaussian mixture does not), and the kernel sparsemax case actually highlights specific peaks and troughs of the original signal, suggesting the ability to attend to higher frequencies. Figure 7 shows rescaled kernel softmax attention densities along with the original series. These are generally smooth and show rich multimodality, and often highlight $0$ crossings where the signal decreases. Figure 9 shows the same for kernel sparsemax. These are very interesting, and learn rich sparsity patterns and often highlight the peaks and troughs of the series, while taking $0$ values over many regions where the series becomes negative.

This dataset uses the architecture in Fig. 11, further described in Appendix I. Hyperparameters are in Table 7. On this dataset, claiming interpretability is difficult because it is engine noise while the class meanings are unknown: only the binary labels are present. However it is interesting that kernel sparsemax is able to select individual waves and exhibit rich sparsity patterns, while kernel softmax highlights $0$ crossings.

| *Hyperparameter* | *Value* |
|---|---|
| Batch Size | 64 |
| LSTM Hidden Units | 128 |
| Value Basis Functions | 256 |
| Inducing Points | 500 |
| Integration Grid Points | 500 |
| Learning Rate | 1e-3 |
| Weight Decay | 1e-5 |
| Epochs | 100 |

Table 7: Hyperparameters for FordA Experiment

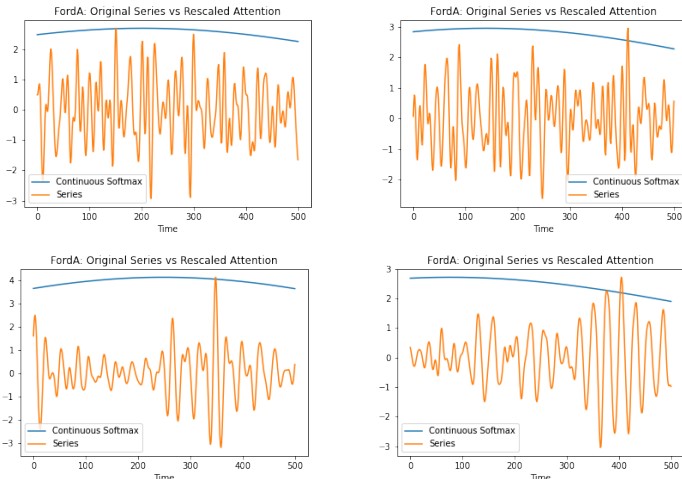

Figure 4: Original time series vs rescaled attention densities for four randomly selected examples from the FordA dataset using continuous softmax. The densities are rescaled so that they have the same max as the signal. The densities are simple and do not attend to fine portions of the signal.

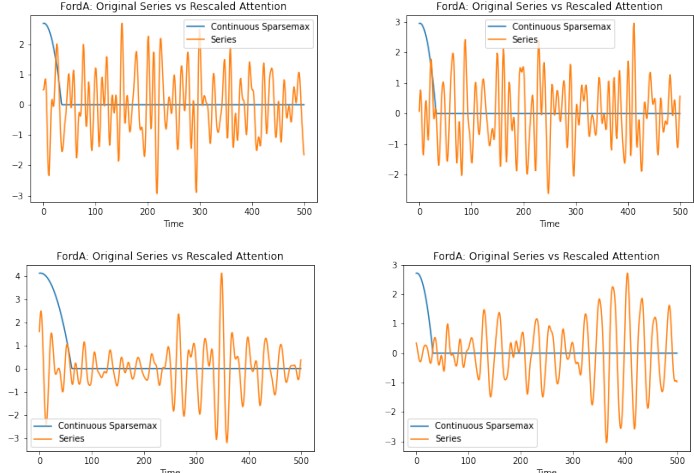

Figure 5: Original time series vs rescaled attention densities for four randomly selected examples from the FordA dataset using continuous sparsemax. The densities are rescaled so that they have the same max as the signal. The densities are again simple and do not attend to fine portions of the signal, although have more focus than in the continuous softmax case.

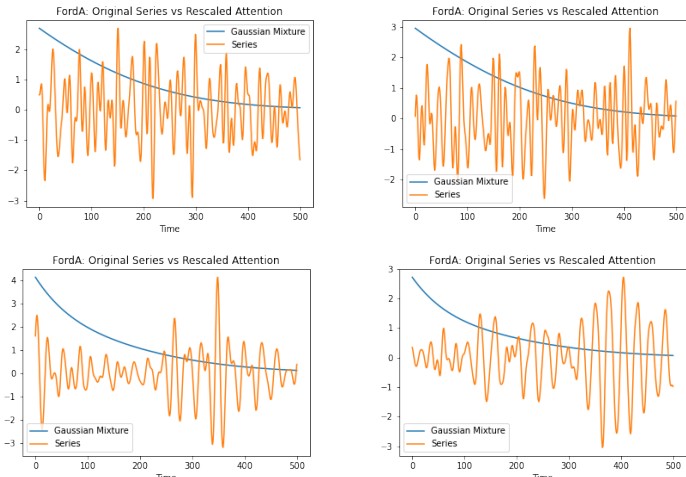

Figure 6: Original time series vs rescaled attention densities for four randomly selected examples from the FordA dataset using Gaussian mixture. The densities are rescaled so that they have the same max as the signal. The densities do not appear multimodal, again likely due to lack of separation between components. However, the shape looks different from the simple shapes of the Gaussian and truncated parabola from continuous softmax and sparsemax.

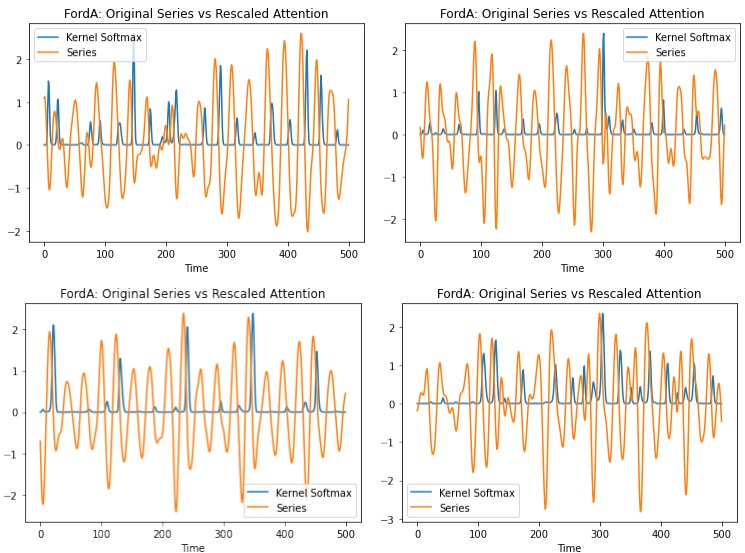

Figure 7: Original time series vs rescaled attention densities for four randomly selected examples from the FordA dataset using kernel softmax. The densities are rescaled so that they have the same max as the signal. The densities seem to focus often on the zero crossings where the signal moves from a peak to a trough.

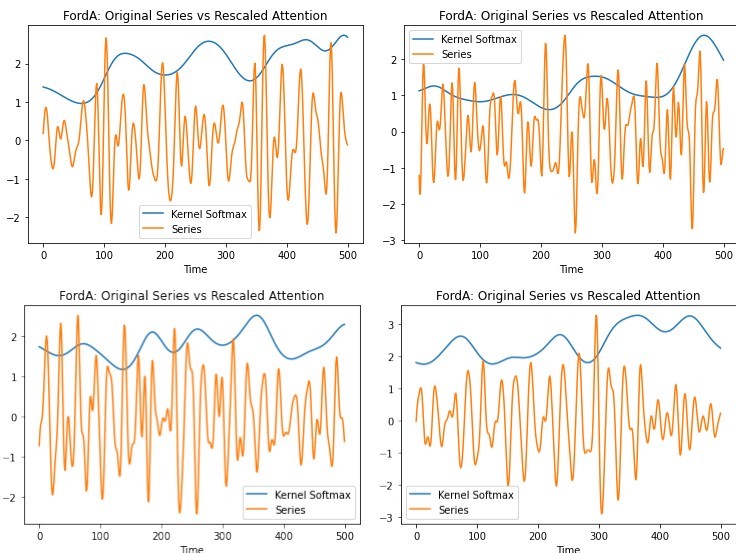

Figure 8: Original time series vs rescaled attention densities for four randomly selected examples from the FordA dataset using kernel softmax. This used a Titan X and a deprecated Pytorch version, but is otherwise the same model as 7, which used Pytorch 1.12 and an A40. The densities are rescaled so that they have the same max as the signal. The densities show a rich multimodal pattern, but do not appear to highlight obvious features of the signal.

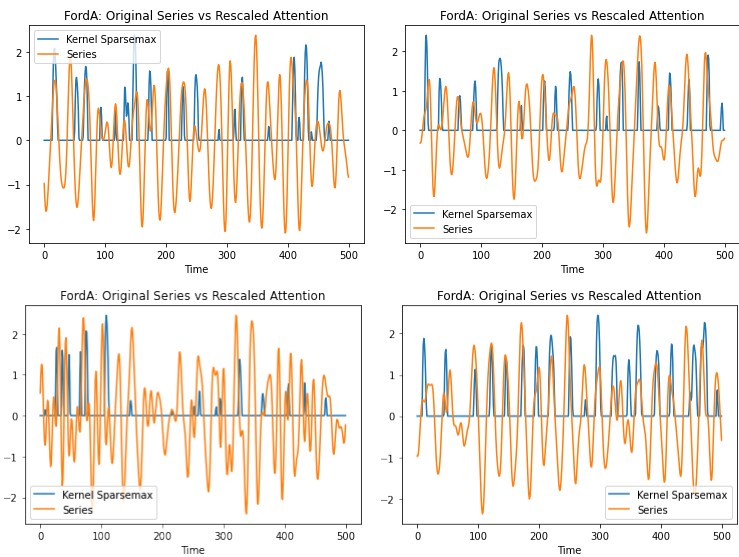

Figure 9: Original time series vs rescaled attention densities for four randomly selected examples from the FordA dataset using kernel sparsemax. The densities are rescaled so that they have the same max as the signal. The densities exhibit very similar patterns to the signal itself, often selecting the peaks or troughs. They also have rich sparsity patterns.

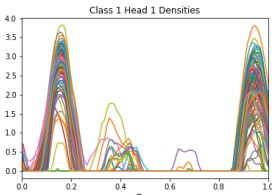 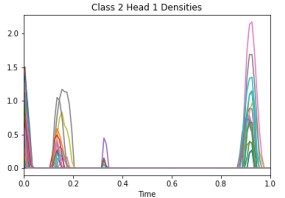 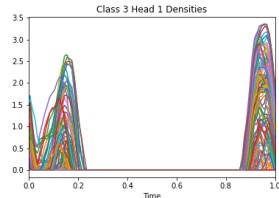

Figure 10: Attention densities for head 1 for different classes for the uWave dataset. We can see that within each class, the densities tend to be similar, while between classes they are less so, suggesting the densities learn to represent classes.

| Hyperparameter | Value |
|---|---|
| Batch Size | 25 |
| Value Basis Functions | 64, 128 |
| Heads | 100 |
| Inducing Points | 256 |
| Integration Grid Points | 100 |
| Optimizer | Adam |
| Learning Rate | 1e-4 |
| Epochs | 10 |

Table 8: Hyperparameters for uWave Experiment

# H    Additional uWave Details

We experiment with $N = 64, 128$ basis functions, and use a learning rate of $1e - 4$. We use $H = 100$ attention mechanisms, or heads. Unlike [42], our use of multiple heads is slightly different as we use the same value function for each head, and only vary the attention densities. Additional architectural details are given below. Table 8 summarizes the hypermarameters and training details.

## H.1    Value Function

The value function uses regularized linear regression on the original time series observed at random observation times (which are not dependent on the data) to obtain an approximation $V(t; \mathbf{B}) = \mathbf{B}\Psi(t) \approx X(t)$. The $H$ in Eqn. 6 is the original time series.

### H.1.1    Encoder

In the encoder, we use the value function to interpolate the irregularly sampled time series at the original points. This is then passed through a convolutional layer with $4$ filters and filter size $5$ followed by a max pooling layer with pool size 2. This is followed by one hidden layer with $256$ units and an output $v$ of size 256. The attention densities for each head $h = 1, \ldots, H$ are then

$$\mu_h = w_{h,1}^T v$$
$$\sigma_h = \text{softplus}(w_{h,2}^T v)$$
$$\gamma_h = W^{(h)} v$$

for vectors $w_{h,1}, w_{h,2}$ and matrices $W^h$ and heads $h = 1, \ldots, H$

### H.1.2    Attention Mechanism

After forming densities and normalizing, we have densities $p_1(t), \ldots, p_H(t)$, which we use to compute context scalars

$$c_h = \mathbb{E}_{p_h}[V(T)]$$

We compute these expectations using numerical integration to compute basis function expectations $\mathbb{E}_{p_h}[\psi_n(T)]$ and a parametrized value function $V(t) = B\psi(t)$ as described in section 3.

| Method | Accuracy | F1 |
|---|---|---|
| Discrete Softmax | 97.97 | 88.67 |
| Cts Softmax | 98.20 | 91.01 |
| Cts Sparsemax | 98.30 | 90.92 |
| Gaussian Mixture | 98.12 | 90.16 |
| Kernel Softmax (ours) | **98.67** | **92.56** |
| Kernel Sparsemax (ours) | **98.42** | **92.07** |
| LSTM FCN | 98.36 | 90.45 |
| TST (Transformer) | 98.18 | 90.79 |

Table 9: Accuracy results on MIT BIH Arrhythmia Classification dataset. The first four rows use the same architecture but different attention mechanisms. Gaussian mixture had 10 components: other choices were tried with lower performance. Rows five and six are our attention mechanisms. LSTM FCN [13] is an LSTM+fully convolutional network. TST is a transformer from [46]

### H.1.3 Classifier

The classifier takes as input the concatenated context scalars as a vector. A linear layer is then followed by a softmax activation to output class probabilities.

## I ECG Heartbeat Classification: MIT BIH

We use a kaggle version[8] of the MIT Arrhythmia Database [9]. Not all versions are comparable: [12] report results under well-balanced classes (unclear how they obtain these), while [26] augment the dataset with SMOTE. We do no data augmentation, so compare to two time series classification baselines in TSAI: a hybrid fully convolutional and LSTM network[13], and a transformer [46].

The task is to detect abnormal heart beats from ECG. The five classes are {Normal, Supraventricular premature, Premature ventricular contraction, Fusion of ventricular and normal, Unclassifiable}. There data has a 87,553/21,891 train/test split. Each sample is a univariate time series of length 187: we pass this through two convolutional layers in order to obtain a multivariate representation of each time step. We then use an LSTM. The hidden layer is used to construct discrete attention using [3]. Following [24] for unimodal continuous softmax and sparsemax, we first output discrete attention weights $p = (p_1, \ldots, p_L), p \in \Delta^L$ in the probability simplex and then compute $\mu = \mathbb{E}_p[T/L]$ and $\sigma^2 = \mathbb{E}_p[(T/L)^2] - \mu^2$ where $T \sim p$. The value function uses Gaussian RBFs. The final context vector is passed through three feedforward layers. Kernel softmax attention is not particularly interpretable. For kernel sparsemax, the attention densities tend to highlight peaks in the signal. Particularly they assign high weight to the R wave, the peak of the QRS complex, of the heartbeat.

We plot attention densities for randomly selected examples under the different density classes, and find that only kernel deformed exponential families/sparsemax learn interpretable attention densities, which focus on regions where the electrical signals from the heart are strong. Figures 12 and 13 show attention densities vs original signals for continuous softmax and sparsemax, respectively. Both are only able to learn simple unimodal densities. Figure 14 shows the same for Gaussian mixture attention. These do not look very multimodal, despite having one component per time point. This is likely due to lack of separation between components. However, we do see that the shapes look more flexible than in single Gaussian or truncated parabola cases. Figure 15 shows attention densities vs original signals for kernel softmax attention. While not particularly interpretable, it learns densities similar to exponential densities without them being specified, a benefit of being a non-parametric density. Figure 16 shows the same for kernel sparsemax. These show interesting highlighting of waves, which describe electrical signals passing through the heart conduction system. There is a particular focus on the R wave, the largest peak in a heartbeat, representing electrical stimulus in the main ventricular mass.

Note that given the relatively complex model structure and reasonably high capacity relative to the original time series length (512 LSTM hidden units per time step for a univariate time series of length 187), models without an interpretable attention density may still perform well by taking advantage

---

[8]https://www.kaggle.com/datasets/shayanfazeli/heartbeat, license Open Data Commons Attribution License v1.0

| Hyperparameter | Value |
| --- | --- |
| Batch Size | 64 |
| LSTM Hidden Units | 512 |
| Value Basis Functions | 24 |
| Inducing Points | 187 |
| Integration Grid Points | 187 |
| Learning Rate | 1e-3 |
| Weight Decay | 1e-5 |
| Epochs | 30 |

Table 10: Hyperparameters for MIT BIH Experiment

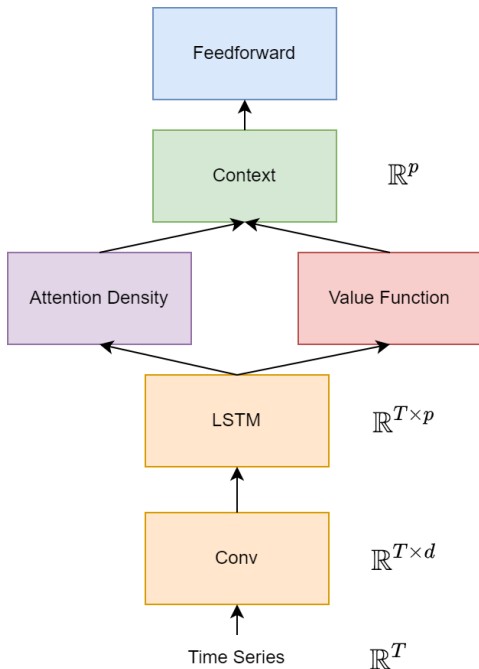

Figure 11: Architecture used for the MIT BIH and FordA experiments. The input is a univariate time series. The conv block has two conv layers, with the goal of converting a univariate time series into a multivariate one. The hidden units of the LSTM are then used to compute the attention density parameters and the value function. The feedforward network has three layers, where the first two have ReLU activation functions.

of capacity elsewhere in the model. However, a model that selects ECG waves may be useful in convincing specialists of its value.

## I.1 General Architecture

Our general architecture is shown in 11. In the first part, two convolutional layers of filter size 5 and 24 filters with padding map the original univariate time series from $\mathbb{R}^{187}$ to a multivariate representation $\mathbb{R}^{187 \times 24}$. This is then passed to an LSTM. Context vectors are computed using either the original hidden states (discrete attention) or a continous-time representation (continuous attention). The context vector is then fed into a feedforward network for final classification. The entire architecture is trained end-to-end. Hyperparameters are described in Table 10.

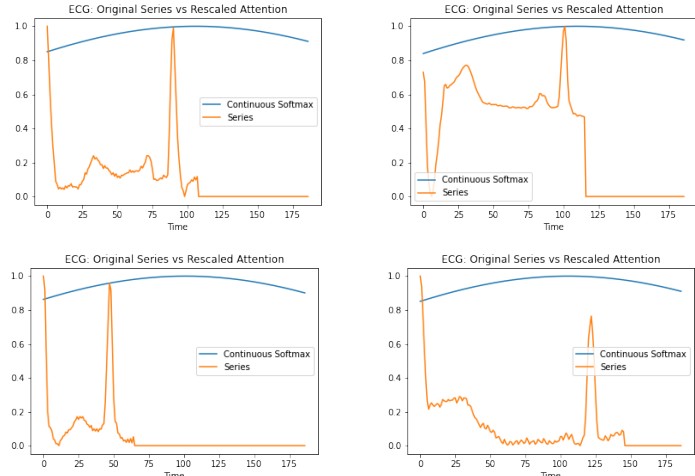

Figure 12: Original time series vs rescaled continuous softmax attention densities for four randomly selected examples from the MIT BIH dataset. Continuous softmax only learns simple unimodal densities.

## I.2 Value Function

The value function uses regularized linear regression on the hidden states of an LSTM to obtain an approximation $V(t; \mathbf{B}) = \mathbf{B}\Psi(t) \approx h_t$. The $H$ in Eqn. 6 is the set of all hidden states.

## I.3 Encoder

The encoder takes the hidden layer of the LSTM as input. For the hidden units $h_t \in \mathbb{R}^p$ for a given time step, it computes

$$v_t = \tanh(W_w h_t + b_w), W_w \in \mathbb{R}^{d \times p}, b_w \in \mathbb{R}^p$$
$$\gamma_t = w_v^T v_t$$

Note that this is written slightly differently from the form in the main paper.

## I.4 Attention Mechanism

The attention mechanism takes the parameters from the encoder and forms an attention density. It then computes

$$c = \mathbb{E}_p[V(T)]$$

for input to the classifier.

## I.5 Classifier

The classifier has three feedforward layers, where the first two have ReLU activation functions.

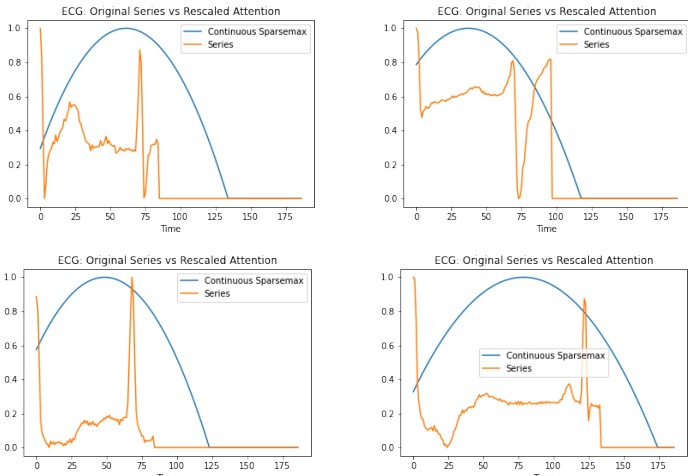

Figure 13: Original time series vs rescaled continuous sparsemax attention densities for four randomly selected examples from the MIT BIH dataset. Continuous sparsemax, similar to continuous softmax, again only learns simple unimodal densities.

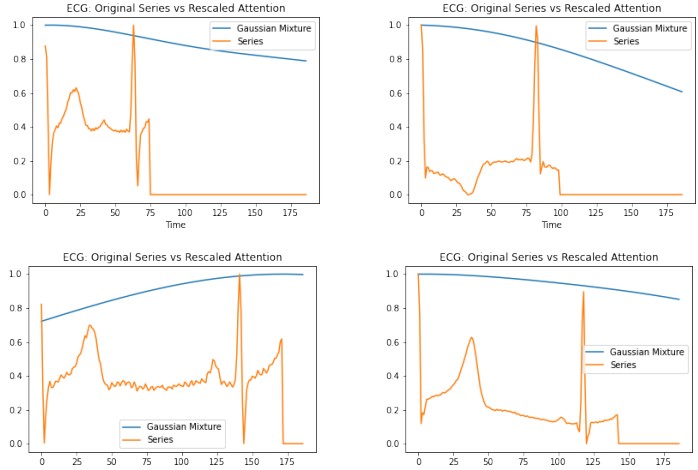

Figure 14: Original time series vs rescaled Gaussian mixture attention densities for four randomly selected examples from the MIT BIH dataset. Number of mixture components equal to number of time points. This does not appear very multimodal, likely because the mixture components are not well separated. However, the attention densities seem to have more flexible shapes than for continuous softmax/sparsemax.

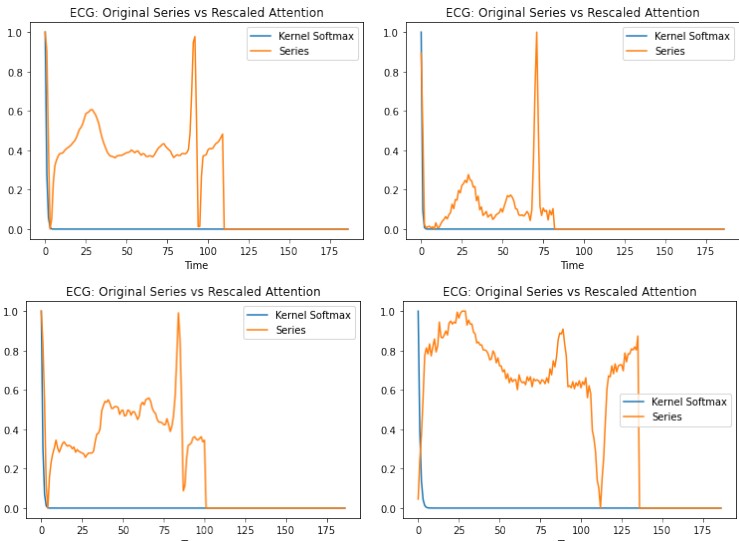

Figure 15: Original time series vs rescaled kernel softmax attention densities for four randomly selected examples from the MIT BIH dataset. While this is not particularly interpretable, it still learns a density that looks similar to an exponential density without explicitly specifying this shape. Further, the empirical performance beats the other attention mechanisms other than kernel sparsemax.

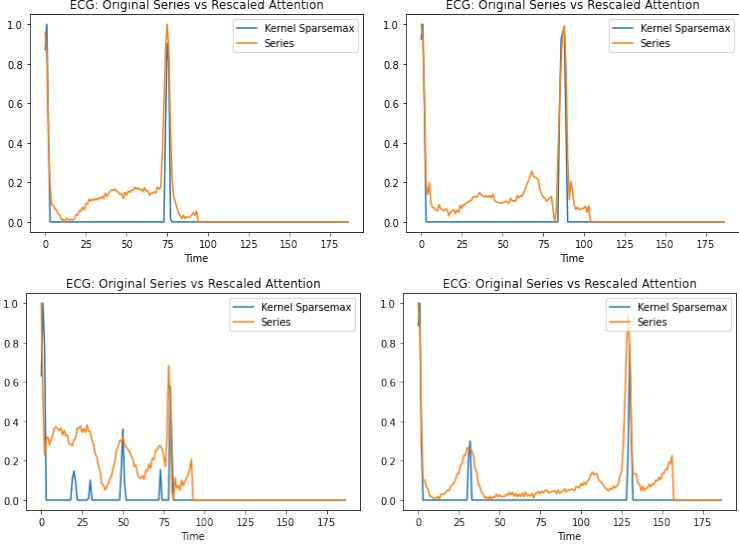

Figure 16: Original time series vs rescaled kernel sparsemax attention densities for four randomly selected examples from the MIT BIH dataset. All cases are multimodal. Further, this is much more interpretable than any of the other methods, and tends to select local peaks, or waves. These waves represent the electrical stimulus as they pass through different parts of the heart conduction system. There is a particular focus on the R wave, the largest peak, which represents the electrical stimulus in the main ventricular mass.