# OpenReview forum: "Kernel Multimodal Continuous Attention"
_NeurIPS.cc/2022/Conference — NeurIPS 2022 Accept_

### Official Review · Reviewer_rxaS · 2022-07-06

**Rating:** 7
**Confidence:** 3
**Soundness:** 3 good
**Presentation:** 3 good
**Contribution:** 3 good

**Summary:**

The authors provide sufficient conditions under which the normalising constant of a kernel exponential family exists (proposition 5.1). Furthermore, they introduce so-called deformed kernel exponential families (sec 5.2) and provide a similar normalisation theory (corollary 5.2), and show conditions under which deformed kernel exponential families are dense in a class of deformed exponential families that uses the space of continuous functions vanishing at infinity in place of the RKHS (appendix D.4). The deformed variant allows for modelling (infinite dimensional) densities with finite support (Figure 1).

These technical results, which are of broad interest, are then applied to a continuous version of kernel attention. Continuous kernel attention is framed as an expectation over a density that is the solution to a regularised strictly convex functional (2). In this case, the (deformed) kernel exponential families are used as the space from which the solution is drawn.

The authors try their new versions of kernel attention on some toy datasets, observing better performance than other methods.

**Questions:**

- I did not properly grasp the motivation behind studying time warping. Is because certain multimodal densities can be represented using time warping? Would it be possible to make the motivation more clear at the start of section 4?
- Is it possible to use kernel and deformed kernel exponential families without the normalisation constant, especially in the cases where the normalisation constant does not exist? Would this ever be justified?
- How does your notion of attention compare with the deep learning definition? Especially with respect to "self-attention"? Is this more close to "self-attention" than general attention?

**Limitations:**

(Not that it affects my evaluation of the paper in any way, but) IMO there is no need to include the last sentence in section 6. Hate speech is one of many potential negative societal impacts, and anyone who has read this far into the paper will already appreciate that this is a largely theoretical work without direct societal application in mind. Hate speech is a non-trivial issue. Singling out hate speech without any context earlier in the paper looks lazier than if it had not been mentioned at all, and does not respect the issue.

**Strengths And Weaknesses:**

Strengths:
- I enjoyed this paper. The authors seem to have properly handled normalisation constants for kernel and deformed kernel exponential families. I like that new theory that may be of independent interest is developed. The approach is thought-out and principled, unlike some other attempts to generalise attention.


Weaknesses:
- I did not like the way equation (1) was presented. It was simply stated "It is (1)". Can you provide a reference for this definition? Or is this your definition? Also see question in Question section below.
- In line 235, we use a finite span for a representation of the function in the RKHS. Is this invoking the RKHS representer theorem? It does not apply to the regularised problem (2) because this concerns the density itself not the f \in RKHS. So where does this finite span representation come from? It seems like Algorithm 1 assumes the coefficients (parameters) of the representer are already known, so presumably one way to justify this is by wrapping algorithm 1 in some regularised empirical risk minimisation problem? I am happy with either a mathematically rigourous justification or a handwavy justification, as long as the handwaving is clearly admitted in the paper.

---

> ### Author Response · Authors · 2022-08-01
> **Response to Reviewer rxaS**
>
> Thank you for your positive comments.
>
> Response to Strengths and Weaknesses
> 1. Equation 1 was from Martins et al. (2020) p. 6 right before Definition 2. We now cite it in the revision.
> 2. For density estimation, this form with finite span is indeed the solution to an empirical risk minimization problem (Sriperumbudur et al. 2017). For our setting we aren't claiming any theoretical justification: using this is simply a way to compute and estimate an $f$ in an RKHS practically. We have clarified this point in the revised paper.
>
> Response to Questions
> 1. We treat this in the general comments to all reviewers (see the first section in the rebuttal). We have actually rewritten section 4 to make it clearer. The main motivation is that we might want a summary statistic of curves where features are aligned in time. One such summary statistic is the expectation of the aligned curves with respect to a global density, thus weighting temporal features in the curves. However we only observe unaligned curves. It turns out that the expectation of the aligned curves with respect to a global density is equivalent to the expectation of unaligned curves with respect to an individualized density. However even if the global density is unimodal, the individualized density is multimodal. Thus we need multimodal continuous attention.
> 2. Interesting question. One can never compute $\int_S p(t)V(t)dQ(t)$, since $p(t)$ does not exist. One could however have a situation where $\tilde{p}$ the unnormalized 'density' is not integrable but $\tilde{p}V$ is. While $\int_S \tilde{p}(t)V(t)dQ(t)$ loses much of the interpretation from Martins et al. (2020) as it is no longer an expectation of $V(T)$, it could make sense to use if one finds that it performs better empirically. We will investigate this in future work, thanks for the suggestion.
> 3. This is the same definition as the deep learning definition, with the main sources of variation being how the value function $V$ and attention density/pmf are represented. In most deep learning work $p$ is a categorical distribution where discrete softmax was used to normalize it. In self-attention, $V$ is a linear transformation of a sequence and one uses a conditional expectation, conditional on a specific token. A context vector is then taken conditioned on a token in the sequence, and one context vector is computed for each token in the sequence.
>
> Response to Limitations:
> We have removed the mention of hate speech.

---

> > ### Comment · Reviewer_rxaS · 2022-08-07
> > **response**
> >
> > Thanks for responding to my review. I am satisfied with your response and happy to recommend acceptance.

---

> > > ### Author Response · Authors · 2022-08-09
> > > **Response**
> > >
> > > Thank you for your support!

---

> > > > ### Comment · Reviewer_rxaS · 2022-08-09
> > > > **typo**
> > > >
> > > > I noticed you wrote `prolem' when representing $f$ in an RKHS.

---

> > > > > ### Author Response · Authors · 2022-08-09
> > > > > **typo**
> > > > >
> > > > > Thank you, fixed!

---

### Official Review · Reviewer_W1Fi · 2022-07-09

**Rating:** 5
**Confidence:** 2
**Soundness:** 3 good
**Presentation:** 3 good
**Contribution:** 2 fair

**Summary:**

In this paper, the authors proposed a multimodal attention density based on kernel exponential families and kernel deformed exponential families for the continuous attention mechanisms. Furthermore, the authors performed the theoretical analysis about normalization, approximation capabilities and properties. In addition, the authors conducted a set of experiments to evaluate the performance of the proposed multimodal attention density, the experimental results shown that kernel continuous attention often outperforms unimodal continuous attention.

**Questions:**

1) what is the role of the Time warping section? Is it to prepare for either section 6.1  or to introduce Lemma 4.2?
2) For the proof of Lemma 5.3 in the Appendix, the first equals sign may be difficult to understand for most readers. Please give the definition of α-exponential.
3) There may be one more word in the Related work section. “They only used unimodal (deformed) exponential family densities:”
4) For equation (6), the matrix dimensions may be incorrect.
5) For the experimental results in Table 2 and Table 3, the standard deviation of the results is missing.
6) Additional time and space complexity with wall-clock time reporting will be helpful.


**Strengths And Weaknesses:**

## Strengths:
- The concept of kernel deformed exponential families is proposed, which is a sparse multimodal density class.
- The overall presentation of this paper is good.

## Weaknesses:
1) The motivation of this work is not well organized or elaborated for using multimodal attention density.
2) This paper tends to be an incremental work. The authors applied kernel methods to the (deformed) exponential families to construct a multimodal attention density for continuous attention. However, the kernel exponential families deformed exponential families, and continuous attention is existing works.
3) Experiments are not enough, why use only two performance metrics and a few datasets in the experiments section?

---

> ### Author Response · Authors · 2022-08-02
> **Response to Reviewer W1Fi**
>
> Response to Strengths and Weaknesses
> 1. We have rewritten section 4 to help motivate the need for multimodal continuous attention.
> 2. We address novelty in the response to common comments for all reviewers.
> 3. These are standard classification metrics. We use four real datasets: two sensor datasets, one audio dataset, one NLP dataset. We also use one synthetic dataset. This is a similar standard for experimental evaluation as in Martins et. al.
>
> Questions
> 1. We address the time warping section in the response to common comments for all reviewers. We have also rewritten it in the revised manuscript.
> 2. We have added a reference to the definition in the main paper in that proof.
> 3. Thank you. Fixed.
> 4. Thank you. We were denoting the observation dimension using both $O$ and $D$. We fixed it to use only $D$.
> 5. These results take some time as they require re-running all experiments at least 10 times. However we will include additional confidence intervals for the final revision. We currently have them for both the synthetic experiment and the uWave experiment in the supplementary material, where the datasets are smaller.
> 6. On an A40 GPU, which we recently got access to and did not use in the original experiments, one epoch of training the ECG example takes between 20-25 seconds for each of cts softmax, cts sparsemax, kernel softmax and kernel sparsemax, so all are similar. Note that we originally trained on a Titan X (an older GPU which we do not currently have access to) and we recall much longer training times and a larger difference between the kernel-based vs unimodal attention. We suspect that the A40 has made everything more efficient, and that the differences due to numerical integration are hidden by operations that take longer. We will add these profiling details for all the experiments to the final revision.

---

> > ### Comment · Reviewer_W1Fi · 2022-08-09
> > **Response**
> >
> > Thanks for responding to my review and efforts to improve the paper.

---

> > > ### Author Response · Authors · 2022-08-09
> > > **Response**
> > >
> > > Thank you, we are glad that we were able to clarify your points.

---

### Official Review · Reviewer_oPnf · 2022-07-10

**Rating:** 8
**Confidence:** 4
**Soundness:** 4 excellent
**Presentation:** 3 good
**Contribution:** 4 excellent

**Summary:**

This paper proposes an extension of the continuous attention formalism presented by Martins et al. (2020). This extension consists in replacing the linear parameterization of the score function ($f$) in continuous attention by a kernel in an RKHS, yielding a kernel exponential family density ($\tilde{f}$). Using q-exponentials, the authors propose using _kernel deformed exponential families_, resembling the entmax formulation of Martins et al. (2022) for $1 < \alpha \leq 2$, but with a special form for $f$. The main challenge in this new formulation is that computing Equation 1 with $\tilde{f}$ is harder, and thus the authors resort to numerical integration for the forward pass and rely on automatic differentiation for the backward pass. The authors also show that numerical integration can be done efficiently with exponential convergence for kernel exponential attention. The paper presents three carefully chosen experiments to showcase multimodal continuous attention: Time Warping,  ECG heartbeat classification, and Automotive symptom detection. The results indicate that kernel multimodal continuous attention is very effective, even when compared to standard continuous softmax and sparsemax attention, and it also performs better than Gaussian mixture, which also yields multimodal densities.

**Questions:**

Have you encountered convergence problems with numerical integration?
Which hyperparameters can be tuned to get better optimizations?
How long does it take to run continuous kernel sparsemax compared to continuous sparsemax?


**Limitations:**

The authors have addressed the limitation and potential negative societal impact of their work adequately.

**Strengths And Weaknesses:**

This paper has enough content to be appropriately divided into two main views: practical and theoretical.

On the practical side, the paper does a great job at casting deformed exponential families within the framework of continuous attention of Martins et al. (2020). The experiments were carefully chosen to illustrate the main advantages of the new formulation, mainly due to the necessity of having multimodal densities. On top of that, the authors also compared their work with related works, including the Gaussian Mixture approach by Farinhas et al. (2021). The results of the experiments are impressive as well as expected, given the importance of multimodality. The paper also presents results for text classification in Appendix G, but I believe this experiment could be in the main paper, as it was a key result reported by Martins et al. (2020). A clear downside of this work is that it requires numerical integration for the forward pass (even though the convergence is fast), which might forbid its application in specific applications. Given this discrepancy to unimodal continuous softmax/sparsemax, the paper can be improved by reporting also the runtime alongside test performance.

On the theoretical side, I believe the paper is very dense. Despite being comprehensive, the appendix is overwhelming. Some proofs could point to other works, such as D.1 -> B.3 in Martins et al. (2020). For me, it is not very clear why Proposition 5.1 and Corollary 5.2 is needed in the paper. It seems an orthogonal result to the practical side. Other than this, up to my knowledge, the theoretical aspect of the paper looks sharp.

Overall, I believe the authors introduced the problem and the motivation for having multimodal continuous attention very well. The related works section is also well written. I believe that a plot from Figure 9 would further clarify the main point of the paper if it was presented in the introduction (alongside its counterpart plots in Figures 5, 6, 7, 8).

---

> ### Author Response · Authors · 2022-08-02
> **Response to Reviewer oPnf**
>
> Response to Strengths And Weaknesses:
>
> Thank you for your positive comments. We will include a plot from Figure $9$ to help clarify the motivation and cite the relevant proof sections in Martins et al. (2020). We will also move the NLP experiment to the main paper and the engine noise experiment to the appendix. Regarding the existence/normalization results, these show that our choices of $Q$ the base measure and $\mathcal{H}$ the RKHS are valid, and also could inform those who would wish to try other choices of $Q$ and $\mathcal{H}$ to see if they perform well empirically on their datasets/problems. We report runtime results below, which we will include in the final version.
>
> Response to Questions
>
> 1. We noticed two numerical stability issues.
>   a) recently we tried using the default settings on a newer GPU (A40) which uses lower precision than the FP32. Predictive accuracy was drastically worse and attention densities in some cases looked noisier and in some cases collapsed to (nearly) point masses. We thus recommend using FP32 precision.
>   b) we sometimes encountered underflow issues when computing $1/Z$. For kernel exponential families we can use the same standard normalization trick from discrete softmax. Noting that $\exp(f(t)-A(f))=\exp(f(t)-C+C-A(f))$, instead of normalizing $\exp(f(t))$ with $\exp(-A(f))$ we can normalize $\exp(f(t)-C)$ with $\exp(C-A(f))$. For kernel deformed exponential families, it uses the positive part/RELU function and sometimes what is commonly known as 'neuron death' occurs, where the terms in the deformed exponential are negative enough that the positive part outputs $0$ for all of the numerical integration points. When this occurs and leads to NaNs there are two options. The first is to restart training. The second is to force the resulting NaNs to some value such as $0$ and continue training. As long as this stops occurring by the end of training, which it did in all of our reported results, the final estimates are valid and will involve properly computing kernel continuous attention with numerical integration. We will make this explicit in the paper, and search for a more principled approach to this numerical stability issue in future work. Note that this did not occur for most of the experiments, but did occur for the ECG experiment.
> 2. There are four major hyperparameters that we consider part of (kernel) continuous attention rather than part of the full architecture (including the LSTMs and feedforward layers). These are the number of inducing points, the number of basis functions to estimate the value function, the regularization strength for estimating the value function and the number of numerical integration grid points. Additionally, one can vary the choice of $q$ and the RKHS $\mathcal{H}$.
> 3. On an A40 GPU, which we recently got access to and did not use in the original experiments, one epoch of training the ECG example takes between 20-25 seconds for each of cts softmax, cts sparsemax, kernel softmax and kernel sparsemax, so all are similar. Note that we originally trained on a Titan X (an older GPU which we do not currently have access to) and we recall much longer training times and a larger difference between the kernel-based vs unimodal attention. We suspect that the A40 has made everything more efficient, and that the differences due to numerical integration are hidden by operations that take longer. We will add these profiling details for all the experiments to the final revision.

---

> > ### Comment · Reviewer_oPnf · 2022-08-07
> > **Comment**
> >
> > Thanks for your responses and efforts to improve the paper
> >
> > The first question shows a clear (acknowledged) limitation, which can hinder practical problems if not appropriately addressed. So it is important not just to acknowledge it but also to describe it in detail so that future practitioners can avoid the same pitfalls; I believe a succinct description in the appendix is enough.
> >
> > On the same topic, have you considered using implicit differentiation for computing the backward pass w.r.t. $Z$?

---

> > > ### Author Response · Authors · 2022-08-09
> > > **Response to Comment**
> > >
> > > Thank you for your comment on the limitation. We have actually found a solution to the neuron death issue in the deformed exponential case. The issue is briefly mentioned in section 5.3 now, which points to discussion and derivation in Appendix E.1 in the latest revised paper. Essentially we can apply a very similar idea to the standard stabilization technique of discrete softmax, although the reasoning for why it is valid is slightly more involved and requires exploiting properties of deformed logarithms and Lemma 5.3. We will also look at applying implicit differentiation for the backward pass but currently the solution in Appendix E.1 works in practice, as we tried it on the ECG dataset and we no longer have to either restart training or temporarily force the output to some value in a heuristic way.
> > >
> > > For the issue of floating point precision, this limitation is still present, and we will add a short discussion in the appendix as you suggested.

---

### Official Review · Reviewer_PeFu · 2022-07-11

**Rating:** 4
**Confidence:** 2
**Soundness:** 2 fair
**Presentation:** 2 fair
**Contribution:** 2 fair

**Summary:**

The paper generalizes the recent results (Martins et al. 2020, 2021) on continuous attention mechanisms to sparse multimodal density classes, which referred to as kernel deformed exponential families. The authors analyze theoretically three main aspects of the density families: normalization, approximation ability, and evaluation of context function. Then the authors apply the methods to real data experiments.

**Questions:**

1. Line 84-87: Please specify the function domain and image space of V. My understanding is that V itself is a fixed function and has nothing to do with the probability. When S is a discrete set, \{V(t): t\in S\} is a sequence of values over S. You would better not use time series here to call V(t) here since traditional time series refer to a sequence of random variables across time.
2. Line 88-90: Similar to the above comment, since the randomness comes from T variable, this is different from the traditional definition of stochastic processes.
3. Line 93: Please give a proper reasoning of using the notation \mathcal{M}_+^1.
4. Line 95: Please give the definition of a proper functional.
5. Equation (2): Please give the definition of the L2 inner product over measure Q.
6. Line 98: Please give the definition of negative differential entropy.
7. Equation 3: How can you reach equation 3 (you maybe can add a reference here)? What is A(f) here? Is it the normalizing constant?
8. Equation 6: What is norm F here? Is this the Frobenius norm?
9. Line 121-122: The sentence "Physical time may not ...  an individualized time scale transforming physical time" is sort of misleading. Do you want to emphasize the importance of investigating individual level of physical times where a time warping technique is called for?
10. Definition 4.1: Can you give a more intuitive description of the time warping function? You can use some figures or examples here. Do you need t_{ij} to be an increasing sequence over j? Given the third condition for h_i (the equation under line 131), how can you make h_i strictly increasing?
11. Lemma 4.2: Please introduce U before using in equation 7. Also in appendix, B.1, should be proof of "Lemma" 4.2.
12. Section 5.2.2: Please include at least the main theorem in the main text. Otherwise, one has to treat these results in appendix less important.
13. Section 5.3: The presentation in this section is far from clear. Please revise and give more details and explanation.
14. Line 316: typo "to use".
15. Line 77: typo "improved".
16. Line 78-79: I do not get point 5 here. Also the authors write two separate sections on their contributions (at the end of Section 1 and 2). Please revise to make these more concise and clear.







**Limitations:**

Yes. Also see weaknesses section.

**Strengths And Weaknesses:**

The paper extends previous results on continuous attention mechanisms to kernel deformed exponential families. The authors make a great effort to apply their methods in practice. My major concerns are as follows: (1) There are many notation and terminologies used without reference or definition. (2) Given the previous results such as Martins et al. (2020,2021,2022), Farinhas etal. (2021), the theoretical novelty is limited. (3) Since numerical integration has to be implemented in practice and no reliable theoretical analysis is given, the empirical performance of the framework is questionable.

---

> ### Author Response · Authors · 2022-08-01
> **Response to Reviewer PeFu**
>
> Thank you for pointing out several clarity and notation issues. We have revised the paper based on your comments and questions, and provide specific responses below.
>
> Response to Strengths And Weaknesses:
> 1. We have updated the paper based on this comment to address this, keeping in mind your questions on notation, which we address below.
> 2. We discuss novelty in the main comment section above for all reviewers.
> 3. We have this. We prove exponential convergence for the 1d case under smoothness assumptions. We then show that for our choices of $q,V,p$, these smoothness assumptions hold. This is discussed in section 5.3.1 as well as Appendix E.
>
> Response to Questions:
> 1. $V:S\rightarrow \mathbb{R}^D$. For the time series, we are considering $V(t)$ as a realization of a discrete time stochastic process, which is the standard definition of a time series, rather than the stochastic process itself.
> 2. This is the time integral of a realization of a stochastic process with respect to a probability measure, rather than the time integral of a stochastic process with respect to a probability measure.
> 3. We borrow this notation from Martins et al. (2020). We believe that they use this because $p$ is the Radon Nikodym derivative of a positive measure where the measure of $S$ is $1$.
> 4. Line 95: A proper convex functional is greater than negative infinity for all points in the domain, and less than infinity for some point.
> 5. $\langle p,f\rangle_{L^2(Q)}=\int_S p(t)f(t)dQ(t)=\mathbb{E}_{p(t)\sim T}[f(T)]$. We have rewritten the equation in the paper to use the explicit integral form.
> 6. $\Omega(p)=\int_S p(t)\log p(t)dQ(t)$. We have added this to the paper.
> 7. This was proved in Appendix A of Martins et al. (2020). We now cite it in the revised version. It is very similar to the proof that the exponential family maximizes entropy, given expected sufficient statistics.
> 8. Yes, it is Frobenius norm. We now mention that.
> 9. Yes. We also discuss this in the main comment section for all reviewers, and have rewritten the time warping section of the paper.
> 10. We have made comments for this in the general section for reviewers and also rewritten the time warping section in the main paper. The goal of the time warping function is to align curves to have specific features occur at a set of common reference times. Regarding $t_{ik}$ yes we assume that it is increasing with $k$, which we mention in the revised paper. For the monotonically increasing constraint on $h_i$, one way to think about it is that a personalized time can speed up or slow down physical time, but it can't go backwards in time. So for instance you can make this Winter shorter or longer, but you can't go back to the previous Spring. In our setting we don't explicitly estimate $h_i$ or $g_i$, but when they are estimated they are generally assumed to be of the form $C_0+C_1 \int_0^t \exp(W(s))ds$ for some $C_0,C_1>0$ and function $W(s)$. This makes both monotonically increasing so that time cannot go backwards.
> 11. Thank you. Noted.
> 12. Thank you. Noted.
> 13. We have updated section 5.3. It now starts with the high-level goal of the algorithm, then mentions the algorithm first. The remaining text describes what the different parts of the algorithm do.
> 14. Thank you. Noted.
> 15. Thank you. Noted.
> 16. Point $5$ refers to Appendix A. We reference it under 'Attention Mechanisms' in Section $2$.

---

> > ### Comment · Reviewer_PeFu · 2022-08-09
> > **Comment**
> >
> > Thanks for your responses to improve the paper. For the first question, I still do not feel comfortable with the description. I cannot see the point why you choose "time series" to describe the value function. This is misleading especially to the theoretical researchers. If this has been used in previous literature, please cite them.

---

> > > ### Author Response · Authors · 2022-08-09
> > > **Response**
> > >
> > > Thank you for bringing this up. Our intention is to say that the value function can be a time series, but not that it always is. It can also be a realization of a continuous-time stochastic process, a representation of a sentence in NLP, or a representation of an image with image data. These representations can be either raw data or learned. An example where the value function is raw irregularly sampled time series data is [2]. Here for each attention mechanism (eqn. 3 in cited paper), $S$ is the set of observed time points, the value function maps the observed time points to their respective observation values, and the attention probability mass function is the output of normalized kernel evaluations between observation time points and a reference time point. We have added a citation to this in the latest paper revision along with a brief mention that the value function could also be a representation of an image.
> > >
> > > [2] Shukla, Satya Narayan, and Benjamin M. Marlin. "Multi-time attention networks for irregularly sampled time series." arXiv preprint arXiv:2101.10318 (2021).

---

### Author Response · Authors · 2022-08-01
**Response to Common Comments**

We thank the reviewers for their helpful comments and critiques, which have helped us to clarify the paper's presentation. We first address major common comments and then provide individualized responses to each reviewer's comments. For brevity, we respond without quoting the original reviewer questions.

$\textbf{Novelty}$ Following [1], we claim *usefulness* in both modeling properties and application and *technical novelty* in both statements and proof strategies. For useful modeling properties, our proposed kernel deformed exponential families are *multimodal with learned support*. Other common densities have fixed support (Gaussian mixture, kernel exponential) or are unimodal with learned support (deformed exponential families). For multimodal, the support is generally fixed by a hyperparameter (base density kernel softmax/Gaussian mixture). Kernel deformed exponential families have learned support and multimodality, leading to rich sparsity patterns.

For usefulness in application, there are two dimensions. We demonstrate improved prediction performance over baselines on MIT BIH, FordA and uWave datasets, as well as in simulation with unaligned curves based on the time warping motivation, and the attention densities, particularly in the ECG example, tend to highlight meaningful parts of a signal. For ECG, none of the methods outside of kernel deformed exponential families highlight the waves of a signal consistently.

For technical novelty, the normalization results for both kernel exponential and deformed exponential families are new, as are the numerical integration convergence results. We also use a *novel proof strategy* to show approximation results. Previous kernel exponential family results (Sriperumbudur 2017) bound the $L^p$ distance and KL divergence between two Boltzman Gibbs densities (Eqn. 3 in paper) parametrized by $L^\infty$ functions in terms of the $L^\infty$ distance between these functions by applying the product rule for exponentials. We cannot do this for deformed exponentials, but still show an analogous (Lemma D.4) for $L^p$ distance and Bregman divergence. We use a functional mean value inequality and bound individual terms, and also apply Taylor's remainder theorem to powers of densities. Therefore, this paper substantially extends theory of continuous attention models relative to prior work.

$\textbf{Time warping motivation}$ Many curves that are realizations of the same process, such as ECG, have recurring features that are not aligned in time. For instance, the heartbeats in ECG waveforms will not be aligned in time across subjects or recording days, and the hottest and coldest days in a season will not be aligned across geographical locations. We may want a summary statistic of aligned curves that is useful for classifying them. One such statistic is the expectation of each aligned curve with respect to the same global density. This could weight specific features aligned in time, such as heartbeats in ECG or seasonal properties of temperature measurements across different locations. However, we can only observe the unaligned curves. To address this, we show that the expectation of the aligned curve with respect to a global density is equivalent to the expectation of the unaligned curve with respect to an individualized density (Lemma 4.2). However, even for some unimodal choices of global density the individualized density is multimodal (Appendix B.2). This demonstrates that only multimodal continuous attention can represent the desired summary statistic.

$\textbf{Revisions}$: Our main changes to the paper are:
1. We have rewritten section $4$ for time warping to make the motivation clearer.
2. We have improved notation and added some citations/references to equations as well the equations for terms we use words for.
3. We have rewritten the approximation theory section to include a formal statement of the main result.
4. We have rewritten section 5.3, which gives an algorithm for computing kernel deformed exponential family continuous attention.

[1] https://medium.com/@black_51980/novelty-in-science-8f1fd1a0a143

---

### Meta-Review · Area_Chair_rnoC · 2022-08-26

**Recommendation:** Accept
**Confidence:** Certain

**Metareview:**

This is solid contribution overall. The paper is well written and the notation easy-to-follow. The authors spent a lot of effort to address the clarity issues by reviewer PeFu. Requiring numerical integration is a limitation but it is clearly acknowledged in the paper.

We recommend acceptance.

Minor additional remarks:

- The term "value function" was a bit confusing because it has a different meaning in game theory or optimal control.
- Section 4 on time warping does not cite any work. It would be great to connect it better with the existing literature.

**Award:**

No

---

### Decision · Program_Chairs · 2022-09-14

Accept